# MAST: Motif-Augmented Diffusion with Search Tree for Spectroscopic Molecular Structure Elucidation

**Chenghao Jia** [1 2]  **Mengdi Liu** [1 2]  **Hong Chang** [1 2]  **Shiguang Shan** [1 2]  **Xilin Chen** [1 2]

## Abstract

Elucidating molecular structures from spectra is a foundational problem in chemical and materials characterization, yet remains challenging due to spectral ambiguity and the vast molecular space. Although recent diffusion-based generators show strong promise for spectra-conditioned elucidation, existing methods struggle to learn robust spectra-structure relationships from limited paired data when relying solely on global spectral representation. Moreover, the repeated full sampling inference strategy incurs substantial computation overhead. To address these limitations, we propose **MAST**, a **M**otif-**A**ugmented diffusion framework with **S**earch **T**ree, for joint 2D-3D spectroscopic molecular structure elucidation. MAST introduces explicit, interpretable *motif priors* as intermediate evidences throughout denoising, reducing conditional ambiguity and facilitating spectra-conditioned optimization. We further cast diffusion sampling as *reward-guided tree search* to prioritize high-reward denoising trajectories, yielding a compact set of spectra-consistent candidates under limited budgets. On the QM9S multi-spectra benchmark, MAST achieves **94.89%** exact recovery and improves 3D fidelity, while preserving high chemical validity and stability. Code is available at https://github.com/Jia040223/MAST.

## 1. Introduction

Molecular structure elucidation is a long-standing and central problem in chemical and materials characterization, which underpins a wide range of applications, including unknown identification, metabolome annotation, natural-product discovery, and impurity/degradant analysis in drug development (Görög, 2006; Dunn et al., 2013; Coates et al., 2000). Spectra (*e.g.*, Raman, IR and UV-Vis) provide indirect yet informative fingerprints of local chemical environments and bonding patterns, enabling inference of underlying molecular structures. However, molecular elucidation remains underexplored due to the inherently challenging *one-to-many problem*: distinct molecular structures may exhibit very similar spectral signatures. In practice, common spectral-to-structure workflows still rely heavily on spectral library search, expert interpretation, and iterative experimental validation, which limits automation and scalability.

Recent years have witnessed notable attempts for this problem. Early automation largely relied on spectral library matching and rule-driven structure elucidation pipelines, which generate candidate structures and prune them using spectral constraints and heuristic rules (Pesek et al., 2020). With the growth of data scale and learning capacity, model-based approaches were adopted to predict interpretable intermediate cues (*e.g.*, functional groups) from spectra to facilitate downstream reasoning (Fine et al., 2020; Enders et al., 2021). More recently, end-to-end methods made significant progress by directly mapping spectra to molecular representations (SMILES or graphs) through conditional generative frameworks—most notably diffusion models (Alberts et al., 2024; Kanakala et al., 2024; Wang et al., 2025b).

Despite high potential for spectra-conditioned structure generation, current methods still face crucial limitations. *First*, most end-to-end methods condition generation on a single global spectral representation (Alberts et al., 2024; Wang et al., 2025b), which can hardly convey sufficient, discriminative, and chemically meaningful cues. Moreover, it leaves the generator to implicitly recover fine-grained chemical evidence from this compressed representation, making robust spectra-conditioned generation difficult, especially when paired spectra–structure data is limited. *Second*, current inference methods, which rely on repeated full-chain sampling followed by post hoc selection (Cheng et al., 2024; Wang et al., 2025b; Alberts et al., 2025), is computationally expensive and difficult to control. As illustrated in Figure 1, it may repeatedly explore similar trajectories, and even when early intermediates deviate, the entire denoising chain is still

[1]State Key Laboratory of AI Safety, Institute of Computing Technology, Chinese Academy of Sciences, China [2]University of Chinese Academy of Sciences, China. Correspondence to: Hong Chang <changhong@ict.ac.cn>.

*Proceedings of the 43rd International Conference on Machine Learning*, Seoul, South Korea. PMLR 306, 2026. Copyright 2026 by the author(s).

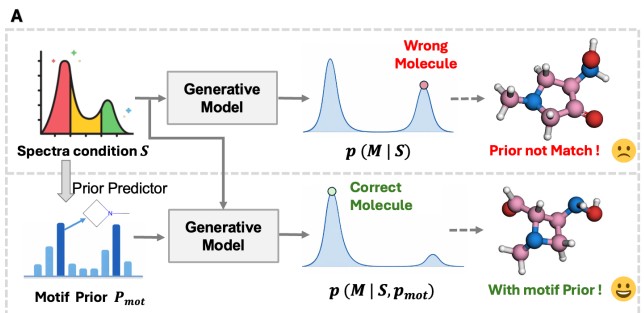
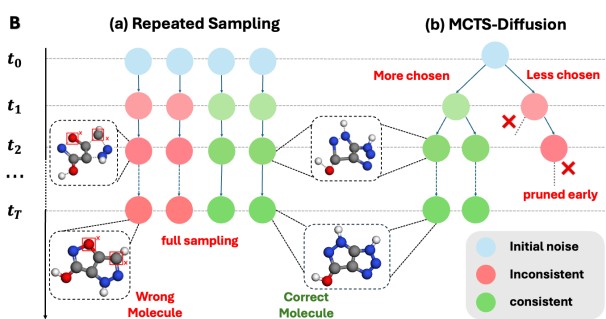

*Figure 1.* **(A)** Motif priors provide interpretable local evidence to filter mismatched candidates and sharpen spectra-consistent generation. **(B)** Repeated sampling explores full reverse-diffusion trajectories, wasting computation on redundant paths and low-consistency prefixes; MCTS-Diffusion performs checkpointed search, expands high-reward intermediates, and prunes low-value branches early.

executed. *Third*, existing works mainly focus on 1D and 2D molecular elucidation, whereas 3D structure is more crucial for many downstream applications. Since 3D recovery is inherently harder due to more complex spectra-geometry relationships and vastly larger search space, direct predicting 3D geometry from spectra remains comparatively scarce.

To address all these limitations, we propose MAST (Motif-Augmented diffusion with Search Tree), a framework for joint 2D-3D molecular elucidation from spectra by leveraging motif-augmented diffusion with tree-search. Specifically, we learn a probabilistic *spectra-to-motif prior* that externalizes implicit spectral cues into interpretable, composable discrete evidence, and inject this evidence throughout joint denoising process to shrink the effective hypothesis space. On the inference side, we cast reverse diffusion as a *segmented decision process* with checkpoints, and introduce a Monte Carlo Tree Search (MCTS) mechanism to adaptively allocate computation between exploration and exploitation—expanding high-reward intermediates while pruning low-value branches early. Finally, we incorporate differentiable conditional guidance at the segment level to adaptively tune guidance strength, thereby improving spectral consistency while preserving the diffusion prior.

Empirically, we conduct comprehensive evaluations on the QM9S (Zou et al., 2023) benchmark. MAST achieves state-of-the-art performance, delivering substantially improved exact recovery while maintaining high chemical validity and stability. Further experiments highlight that the probabilistic motif prior is a key contributor. Moreover, MCTS-guided tree search over diffusion markedly boosts exact recovery under the same denoising-step budget.

In summary, our contributions are as follows:

1. We propose MAST, a motif-augmented diffusion framework for spectroscopic molecular elucidation, with an explicit spectra-to-motif prior as interpretable intermediate evidence.

2. We propose an MCTS-guided diffusion inference algorithm for molecular generation that searches denoising trajectories with segment-wise adaptive guidance.

3. We demonstrate that MAST achieves state-of-the-art performance on QM9S in Top-1 recovery, and that MCTS-guided inference enables *test-time scaling* to improve spectra-consistent recovery under limited sampling budgets *without retraining or fine-tuning*.

## 2. Related Work

**Spectra-to-structure molecular elucidation.** Early spectra-based elucidation primarily relied on expert heuristics and library matching to enumerate and screen candidate structures. (Stein & Scott, 1994; Pesek et al., 2020). Learning methods further infer functional-group and fragment-level cues from spectra for scoring and ranking (Fine et al., 2020; Enders et al., 2021). These retrieval-based methods hinge on the coverage and curation quality of reference libraries, which are often insufficient in practice. Recent work increasingly targets direct generation, from reinforcement-learning graph construction (Ellis et al., 2023) to Transformer-based spectra-to-SMILES translation (Alberts et al., 2024; 2025). More recently, several works also leverage language models to generate SMILES from spectra for structure elucidation (Su et al.; Zhuang et al., 2025). Concurrently, diffusion-based generators have been applied to spectra-conditioned elucidation, supporting both graph-level (Bohde et al., 2025) and geometry-aware (Cheng et al., 2024; Wang et al., 2025b) structure generation, demonstrating strong generative capability. However, diffusion models incur high inference cost because they require sequential denoising, which remains a key practical bottleneck.

**Diffusion-based molecular modeling.** Diffusion models generate data via iterative denoising and have become a standard backbone for molecular generation. Early diffusion formulations for molecules include discrete diffusion on *2D* graphs, which perturbs and denoises categorical node/edge attributes to enable molecular graph generation

(Vignac et al., 2022; Huang et al., 2023a; Vignac et al., 2022; Liu et al., 2024). In *3D* molecular modeling, SE(3)/E(3)-equivariant diffusion provides a principled way to generate or refine coordinates while respecting Euclidean symmetries (Hoogeboom et al., 2022; Xu et al., 2022; 2023). Building on these advances, joint 2D–3D diffusion models unify topology and geometry in denoising process, enabling coherent generation of both bond structures and feasible 3D conformations with better performance (Huang et al., 2023b; Hua et al., 2024; Peng et al., 2023; Vignac et al., 2023).

**Test-time scaling and guided search.** Test-time scaling increases computation at inference to improve outputs without changing model parameters. It has gained significant attention in large language models , including best solution selection (Snell et al., 2024; Wu et al., 2024; Brown et al., 2024), random sampling, self-consistency (Wang et al., 2022) and tree-search methods (Yao et al., 2023; Hao et al., 2023; Zhang et al., 2024; Qi et al., 2024). As a classic tree-search algorithm, MCTS has emerged as a powerful technique for structured exploration in the output space (Yao et al., 2024; Wang et al., 2023; Cheng et al., 2025; Coulom, 2006; Silver et al., 2016; Liu et al., 2025). For diffusion models, guided sampling provides an orthogonal way to steer conditional generation, including classifier guidance and classifier-free guidance (Dhariwal & Nichol, 2021; Ho & Salimans, 2022). Inspired by these advances, we combine MCTS and diffusion guidance, significantly improving spectra-consistent generation under limited budgets.

## 3. Preliminaries

### 3.1. Task Definition

Let $\mathcal{S} = \{s^{(m)}\}_{m=1}^{M}$ denote a set of observed spectra from $M$ modalities, where $s^{(m)} \in \mathbb{R}^{L_m}$ is a 1D signal sampled on a fixed grid for $m$-th modality (e.g., IR, Raman, UV, NMR). A molecule is represented by a *joint* 2D–3D structure

$$\mathcal{M} := (X, V, E), \qquad (1)$$

where $X \in \mathbb{R}^{N \times 3}$ denotes the 3D Cartesian coordinates, $V$ denotes atom-level attributes (node features), and $E$ denotes bond-level attributes (edge features), which together define the molecular graph and its 3D conformation. We aim to learn a conditional generative model $p_\theta(\mathcal{M} \mid \mathcal{S})$ from paired spectra–structure data to capture the inherently one-to-many mapping from spectra to molecular structures.

### 3.2. Diffusion Models

Diffusion models construct a forward noising process that gradually perturbs data until the marginal distribution approaches a tractable prior, and then learn to reverse this process to generate new samples (Sohl-Dickstein et al., 2015; Ho et al., 2020; Song et al., 2020; Kingma et al., 2021). Let

$G_0 \in \mathbb{R}^d$ denote a clean data representation, and $\{G_t\}_{t \in [0,1]}$ denote those in the forward process at different noise levels $t$. A common choice is a Gaussian forward marginal:

$$q(G_t \mid G_0) = \mathcal{N}(G_t \mid \alpha_t G_0, \ \sigma_t^2 I), \qquad (2)$$

where $\alpha_t, \sigma_t \in \mathbb{R}^+$ are time-dependent differentiable functions. They are typically chosen so that the terminal marginal at $t = 1$ satisfies $q(G_1) \approx \mathcal{N}(0, I)$.

The reverse-time generative dynamics can be described by a stochastic differential equation whose drift depends on the gradient (score) $\nabla_{G_t} \log q_t(G_t)$ (Song et al., 2020):

$$dG_t = \left[ f(t)\, G_t - g^2(t)\, \nabla_{G_t} \log q_t(G_t) \right] dt + g(t)\, d\bar{w}_t, \qquad (3)$$

where $\bar{w}_t$ is standard Wiener process and $f(t), g(t)$ are coefficients determined by chosen schedule. The score is usually approximated by a neural network, trained with mean squared error objectives (Ho et al., 2020; Song et al., 2020).

## 4. Method

In this section, we introduce the framework of MAST, as shown in Figure 2. Section 4.1 introduces spectra-derived conditioning evidence, including a global embedding and an interpretable motif prior. Section 4.2 describes the motif-augmented joint 2D–3D diffusion model, which is then reformulated as a diffusion tree. Section 4.3 presents a Monte Carlo Tree Search (MCTS) based inference strategy on the diffusion tree to efficiently identify spectra-consistent structures under limited sampling budgets.

### 4.1. Conditioning Evidence from Spectra

**Global spectral embedding.** We encode multi-modal spectra $\mathcal{S} = \{s^{(m)}\}_{m=1}^{M}$ using a patch-based Transformer encoder $E_{\text{spec}}$ and obtain a global conditioning vector $h_{\text{spec}} = E_{\text{spec}}(\mathcal{S}) \in \mathbb{R}^{d_h}$. Patchification enables the encoder to model the local spectral patterns (e.g., nearby peak shapes), while self-attention over concatenated multi-modal patches captures long-range dependencies and cross-modal relationships at the molecule level. To avoid overfitting to paired spectra–structure supervision, we initialize $E_{\text{spec}}$ self-supervisedly via masked patch reconstruction by optimizing $\mathcal{L}_{\text{rec}} = \mathbb{E}\big[\|\hat{\mathcal{S}} - \mathcal{S}\|_2^2\big]$, and then finetune it jointly with the diffusion denoiser.

**Interpretable motif prior.** Spectra-to-structure inversion is intrinsically underdetermined and a single global spectral embedding is often insufficient to capture the hierarchical mapping from spectra to local chemical environments and then to molecular structure, leaving the model to search in an extremely large molecules space (approximately $10^{60}$ (Polishchuk et al., 2013)). To get out of this dilemma, we intro-

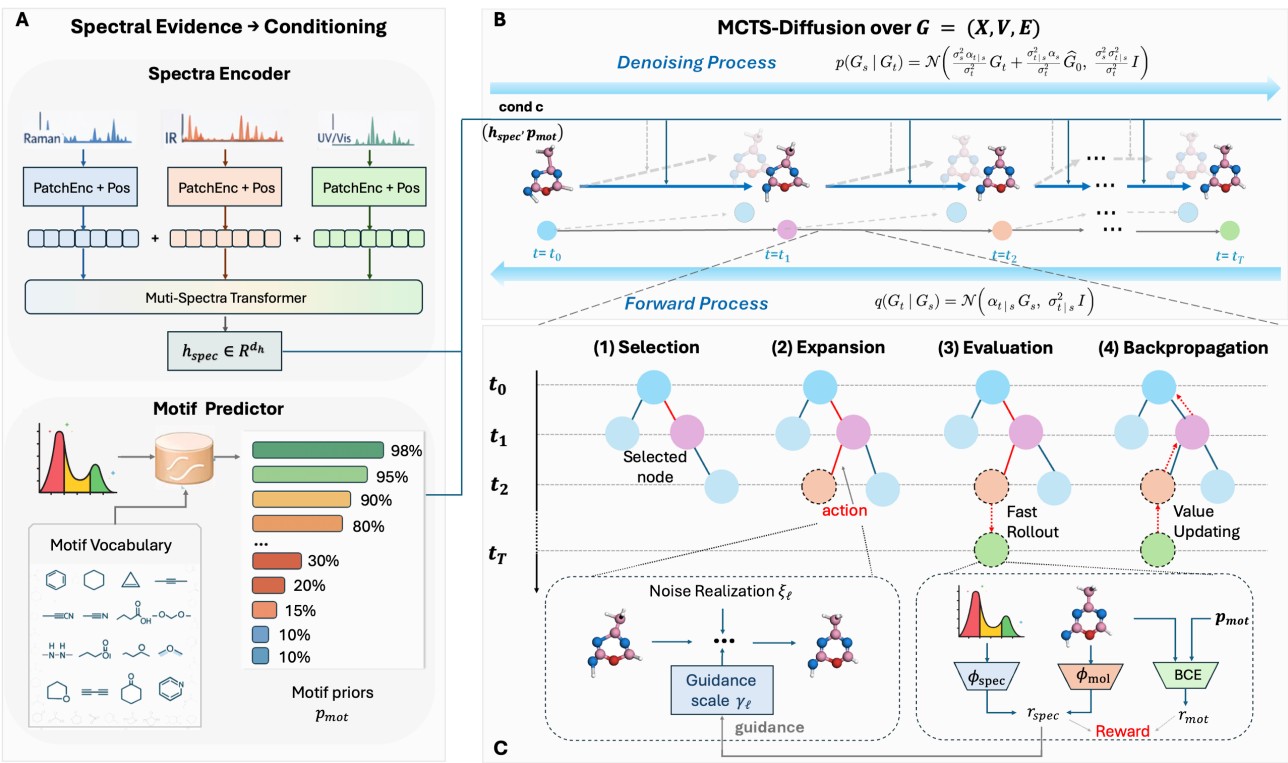

*Figure 2.* Overview of **MAST**. **(A) Spectral evidence**. Multi-spectra encoding produces global context $h_{\text{spec}}$ and a motif prior $p_{\text{mot}}$, yielding $c = (h_{\text{spec}}, p_{\text{mot}})$. **(B) Motif-augmented diffusion as a tree**. Conditioned on $c$, joint 2D–3D diffusion samples the molecular state $G = (X, V, E)$; reverse denoising induces a branching sampling tree of plausible trajectories. **(C) Reward-guided tree search**. Under a limited sampling budget, MCTS prioritizes branches that better match the spectra and the motif evidence.

duce an interpretable *motif prior* that provides motif-level evidence to complement the global embedding.

We construct a vocabulary $\mathcal{V} = \{v_1, \ldots, v_K\}$ with $K=103$ frequent and chemically meaningful motifs. Crucially, this vocabulary covers not only coarse functional group, but a broader set of reusable local motifs, including functional groups and heteroatom-centered local environments, as well as short carbon skeleton fragments, ring systems and common connectivity patterns (see Appendix B for details).

To this end, we train a multi-label predictor $E_{\text{mot}}$ to map spectra to motif presence probabilities. Formally, given a molecule $\mathcal{M}$, we define a multi-hot motif label:

$$y(\mathcal{M}) = (y_1, \ldots, y_K) \in \{0,1\}^K, \ y_k = \mathbb{I}[v_k \subseteq \mathcal{M}]. \tag{4}$$

The motif presence probability is then computed as:

$$p_{\text{mot}} = \sigma(E_{\text{mot}}(\mathcal{S})) \in (0,1)^K, \tag{5}$$

with $\sigma(\cdot)$ the element-wise sigmoid function. $E_{\text{mot}}$ is optimized by a weighted binary cross-entropy (BCE) loss:

$$\mathcal{L}_{\text{mot}} = -\sum_{k=1}^{K} \Big( w_k \, y_k \log p_{\text{mot},k} + (1-y_k) \log(1-p_{\text{mot},k}) \Big). \tag{6}$$

The resulting $p_{\text{mot}}$ is *compositional* and *interpretable*: it provides a calibrated, motif-level evidence profile inferred from spectra. During generation, we encode $p_{\text{mot}}$ as an additional condition and inject it throughout the denoising process, so that the molecular structure could be steered by motif-level evidence, rather than global representation only.

### 4.2. Motif-Augmented Diffusion

We model molecules with a joint diffusion state $G = (X, V, E)$ and learn a conditional denoiser that estimates the reverse diffusion dynamics based on spectral evidence.

**Spectra-conditioned joint denoiser.** We learn an SE(3)-equivariant (w.r.t. 3D rotations and translations) denoiser $D_\theta$ that jointly refines geometry $X$ and graph attributes $(V, E)$ along the reverse diffusion trajectory. We condition denoising on the evidence obtained from Section 4.1:

$$c := (h_{\text{spec}}, p_{\text{mot}}), \tag{7}$$

where $h_{\text{spec}}$ summarizes global spectral context and $p_{\text{mot}}$ provides interpretable motif-level evidence. At each time step $t$, the denoiser $D_\theta$ predicts the clean state $\hat{G}_0$ based on condition $c$ and the current noise state $G_t$:

$$\hat{G}_0 = (\hat{X}_0, \hat{V}_0, \hat{E}_0) = D_\theta(G_t; c, t). \tag{8}$$

Herein, the condition $c$ is injected at every denoising layer via adaptive normalization: it is mapped to layer-wise modulation parameters for node/edge representations. In this way, both global context and motif evidence shape the intermediate message-passing throughout denoising.

**Training objective.** We adopt a variance-preserving diffusion process and train $D_\theta$ in the data-prediction regime (Ho et al., 2020). We sample $t \sim \mathrm{Uniform}[0,1]$, obtain $G_t$ by noising $G_0$, and minimize a weighted reconstruction loss over geometry and topology:

$$\mathcal{L}_{\mathrm{diff}} = \mathbb{E}\Big[\lambda_X\|X-\hat{X}_0\|_2^2 + \lambda_V\|V-\hat{V}_0\|_2^2 + \lambda_E\|E-\hat{E}_0\|_2^2\Big],$$

$$(9)$$

where $\lambda_X$, $\lambda_V$, and $\lambda_E$ control the relative contributions of the geometry and node/edge-attribute reconstruction.

**Checkpointed reverse sampling tree.** The conditional denoiser in Eq. (8), together with a reverse-time sampling algorithm, induces a stochastic reverse-time sampling process for the state $G_t = (X_t, V_t, E_t)$ under spectral evidence $c = (h_{\mathrm{spec}}, p_{\mathrm{mot}})$. Under the same evidence $c$, this stochastic reverse process can realize multiple plausible trajectories, depending on the stochasticity of the sampler and other controllable degrees of freedom. To explicitly represent this trajectory-level variability without modifying the underlying diffusion model, we discretize the *reverse sampling process* into several checkpoints and reformulate it as a tree.

We choose checkpoints $1 = t_0 > t_1 > \cdots > t_L \geq 0$, which define the $L$ levels of the reverse sampling tree. Each node at depth $\ell$ corresponds to a sampled checkpoint state $h_\ell := G_{t_\ell}$. We define a *segment* transition as a block of reverse-time sampling that updates the checkpoint state $h_\ell$ to $h_{\ell+1}$. For each segment, branching arises because of the sampling stochasticity and sampler configurations (e.g. noise trajectory and guidance scale), which we package into an action $a_\ell$. Given an action $a_\ell$, the segment transition from $h_\ell$ to $h_{\ell+1}$ is deterministic:

$$h_{\ell+1} = \mathcal{F}_\theta(h_\ell; a_\ell, c), \qquad (10)$$

where $\mathcal{F}_\theta$ is the segment update induced by denoiser $D_\theta$ under action $a_\ell$. Therefore, a full generation corresponds to a root-to-leaf path $\{(h_\ell, a_\ell)\}_{\ell=0}^{L-1}$ ending at $t_L$. This formulation keeps $D_\theta$ unchanged, while exposing segment-level branching for inference-time tree search.

### 4.3. Reward-guided Tree Search

Repeated full-trajectory diffusion sampling is computationally expensive, as each candidate requires executing the entire denoising trajectory. We therefore perform Monte Carlo Tree Search (MCTS) over the checkpointed reverse sampling tree defined above, prioritizing high-reward prefixes and allocating computation to the most promising branches.

**Spectra-structure reward model.** To guide exploration within the tree, we firstly require a spectra-consistency reward for scoring rollouts. Formally, we embed spectra $\mathcal{S}$ and molecules $\mathcal{M}$ into a shared space:

$$u_{\mathcal{S}} = \phi_{\mathrm{spec}}(\mathcal{S}), \qquad u_{\mathcal{M}} = \phi_{\mathrm{mol}}(\mathcal{M}), \qquad (11)$$

and define the alignment score by cosine similarity

$$\mathrm{Align}(\mathcal{M}, \mathcal{S}) = \frac{\langle u_{\mathcal{M}}, u_{\mathcal{S}} \rangle}{\|u_{\mathcal{M}}\| \cdot \|u_{\mathcal{S}}\|}. \qquad (12)$$

This score serves two roles in our framework: (i) it provides a scalar *reward* for ranking candidate trajectories during tree search, and (ii) it provides a differentiable signal for within-segment gradient guidance.

We apply MCTS to the checkpointed reverse sampling tree defined in Section 4.2. MCTS iteratively performs *selection-expansion-execution-evaluation-backpropagation* to balance exploration and exploitation across candidates.

**Selection.** The selection step starts from the root checkpoint and repeatedly chooses the most promising child to traverse, balancing exploitation of high-value branches and exploration of under-visited branches. Concretely, we compute the upper confidence bound (UCB) for a node $u$ as

$$\mathrm{UCB}(u) = \hat{V}(u) + c_{\mathrm{ucb}}\sqrt{\frac{\log\big(N(\mathrm{pa}(u)) + 1\big)}{N(u) + 1}}, \quad (13)$$

where $N(u)$ is the visit count of node $u$, $\mathrm{pa}(u)$ denotes its parent node, $\hat{V}(u)$ is the estimated node value, and $c_{\mathrm{ucb}} > 0$ controls the exploration–exploitation trade-off. We traverse the tree by repeatedly selecting the child with the highest UCB score until reaching a leaf node.

**Expansion.** After a leaf node at depth $\ell$ with checkpoint latent $h_\ell := G_{t_\ell}$ is selected, we expand it by generating multiple child nodes that correspond to alternative segment-level sampling decisions for the next time interval $[t_\ell, t_{\ell+1}]$. Specifically, we expand the node by proposing $w$ candidate *segment-sampling actions* $\{a_\ell^{(i)}\}_{i=1}^w$, where each action corresponds to a child node and is parameterized by

$$a_\ell^{(i)} = \big(\xi_\ell^{(i)}, \gamma_\ell^{(i)}\big), \qquad i = 1, \ldots, w, \qquad (14)$$

with $\xi_\ell^{(i)}$ specifying the segment-wise noise realization (Gaussian noise draws), and $\gamma_\ell^{(i)}$ controlling the guidance magnitude applied throughout this segment. Executing action $a_\ell^{(i)}$ advances the reverse-time dynamics from $t_\ell$ to $t_{\ell+1}$ under condition $c$ and yields the $i$-th child checkpoint latent $h_{\ell+1}^{(i)} = \mathcal{F}_\theta(h_\ell; a_\ell^{(i)}, c)$. Within the segment, at each internal step from $t$ to $t'$, we perform clean state prediction

*Table 1.* Main structure elucidation results on QM9S (Spectra-to-SMILES baselines are quoted from the original papers; DiffSpectra are reproduced by us under the same evaluation protocol).

| Method | Acc@1↑ | MCES↓ | TaniSimMG↑ | CosSimMG↑ | TaniSimMA↑ | FraggleSim↑ | FGSim↑ | RMSD(Å)↓ | MapAcc↑ |
|---|---|---|---|---|---|---|---|---|---|
| *spectra-to-SMILES models* | | | | | | | | | |
| IR-to-Structure-IR (Alberts et al., 2024) | 0.00% | 11.3187 | 0.0718 | 0.1311 | 0.1585 | 0.1747 | 0.3151 | – | – |
| IR-to-Structure-Raman (Alberts et al., 2024) | 0.00% | 11.3516 | 0.0766 | 0.1395 | 0.1639 | 0.1959 | 0.3525 | – | – |
| IR-to-Structure-UV (Alberts et al., 2024) | 0.00% | 11.4240 | 0.0728 | 0.1326 | 0.1512 | 0.1837 | 0.3151 | – | – |
| Spectra2Structure-IR (Kanakala et al., 2024) | 0.19% | 10.1081 | 0.0965 | 0.1695 | 0.2162 | 0.2308 | 0.4383 | – | – |
| Spectra2Structure-Raman (Kanakala et al., 2024) | 0.00% | 9.4164 | 0.1089 | 0.1901 | 0.2388 | 0.2504 | 0.4419 | – | – |
| Spectra2Structure-UV (Kanakala et al., 2024) | 0.00% | 11.1222 | 0.0716 | 0.1313 | 0.1418 | 0.2092 | 0.3901 | – | – |
| SpectraLLM-IR (Su et al.) | 0.55% | 7.5651 | 0.1921 | 0.3120 | 0.4330 | 0.3194 | 0.6599 | – | – |
| SpectraLLM-Raman (Su et al.) | 3.14% | 6.4076 | 0.2500 | 0.3786 | 0.5071 | 0.2500 | 0.7317 | – | – |
| SpectraLLM-UV (Su et al.) | 0.00% | 10.6374 | 0.0790 | 0.1426 | 0.2026 | 0.2100 | 0.3713 | – | – |
| *Diffusion Models* | | | | | | | | | |
| DiffSpectra (2D&3D) (Wang et al., 2025b) | 60.68% | 1.0514 | 0.7630 | 0.8251 | 0.8609 | 0.8959 | 0.9652 | 0.8091 | 83.84% |
| MAST (w/o MCTS) | 79.12% | 0.6296 | 0.8834 | 0.9141 | 0.9596 | 0.9651 | 0.9806 | 0.8020 | 93.83% |
| MAST (w/ MCTS) | **94.89%** | **0.1048** | **0.9774** | **0.9827** | **0.9908** | **0.9754** | **0.9935** | **0.7843** | **95.34%** |

*Table 2.* Molecular stability and distribution metrics on QM9S (CDGS and JODO are quoted from the original papers; DiffSpectra are reproduced by us under the same evaluation protocol).

| Method | AtomStable↑ | MolStable↑ | V&U↑ | V&U&N ↑ | FCD↓ | SNN↑ | Frag↑ | Scaf↑ |
|---|---|---|---|---|---|---|---|---|
| CDGS (Huang et al., 2023a) | 99.7% | 95.1% | 93.6% | 89.8% | 0.798 | 0.493 | 0.973 | 0.784 |
| JODO (Huang et al., 2023b) | 99.9% | 98.8% | 96.0% | 89.5% | 0.138 | 0.522 | 0.986 | 0.934 |
| DiffSpectra (Wang et al., 2025b) | 99.9% | 98.6% | 96.8% | 90.5% | 0.088 | 0.489 | 0.983 | 0.906 |
| MAST (w/o MCTS) | 99.8% | 98.1% | 97.2% | 89.1% | **0.014** | 0.896 | 0.997 | 0.984 |
| MAST (w/ MCTS) | **99.9%** | **98.8%** | **98.2%** | **97.1%** | 0.044 | **0.981** | **0.999** | **0.997** |

$\hat{G}_0 = D_\theta(G_t; c, t)$ and decode it to a molecule hypothesis $\hat{\mathcal{M}}_0$. We then define an alignment-based guidance loss

$$\mathcal{L}_{\text{guide}}(t) = -\text{Align}(\hat{\mathcal{M}}_0, \mathcal{S}), \qquad (15)$$

to compute $g_t = \nabla_{G_t} \mathcal{L}_{\text{guide}}(t)$, and incorporate the guidance by shifting the reverse transition mean before sampling:

$$G_{t'} = (\bar{G}_{t'} - \gamma_\ell \eta_t g_t) + \sigma_t \epsilon, \qquad \epsilon \sim \mathcal{N}(0, I), \quad (16)$$

where $\bar{G}_{t'}$ and $\sigma_t$ are the mean and noise standard deviation of the unguided reverse-time sampler update for the step $t$ to $t'$, $\eta_t$ is a predefined time-dependent scaling schedule (shared across branches) and $\gamma_\ell$ provides a branch-specific strength multiplier for the current segment.

**Evaluation.** Each expanded node is evaluated by a fast rollout that approximates the final sample quality without running the full reverse trajectory to $t_L$. Given a checkpoint latent $h_\ell$, we perform a *fast rollout* by running the reverse solver for only a few steps to estimate a clean state for evaluation, decode it to a molecule candidate $\tilde{\mathcal{M}}_\ell$, and compute a composite reward with two terms:

$$r(h_\ell) = r_{\text{spec}}(\tilde{\mathcal{M}}_\ell, \mathcal{S}) + \lambda_s r_{\text{mot}}(\tilde{\mathcal{M}}_\ell, p_{\text{mot}}),$$
$$r_{\text{spec}}(\tilde{\mathcal{M}}_\ell, \mathcal{S}) = \text{Align}(\tilde{\mathcal{M}}_\ell, \mathcal{S}), \qquad (17)$$
$$r_{\text{mot}}(\tilde{\mathcal{M}}_\ell, p_{\text{mot}}) = \exp(-\text{BCE}(y(\tilde{\mathcal{M}}_\ell), p_{\text{mot}})).$$

where BCE denotes the binary cross-entropy and $\lambda_s$ is a hyperparameter that controls the trade-off between the spectra–structure alignment and the motif-consistency. If $\tilde{\mathcal{M}}_\ell$ is invalid, we downweight it by a chosen factor $\alpha \in [0, 1)$.

**Backpropagation.** After evaluating newly expanded children, we backpropagate their rollout rewards along the traversal path to update the visit counts and node values, so that future selection increasingly favors high-reward prefixes. We use a *softmax* backup to update the node values, which upweights high-reward rollouts compared to the mean backup, making value propagation more robust to noisy rollouts and improving search efficiency. Given rollout rewards $\{r_i\}_{i=1}^n$ collected at node $u$, we define

$$\hat{V}(u) = \sum_{i=1}^n w_i r_i, \quad w_i = \frac{\exp\left((r_i - r_{\max})/\tau\right)}{\sum_{j=1}^n \exp\left((r_j - r_{\max})/\tau\right)}, \qquad (18)$$

where $r_{\max} = \max_i r_i$ is used for numerical stability in the softmax exponent, $\tau > 0$ is the temperature parameter and $n$ is the visit count $N(u)$ of the node $u$.

## 5. Experiments

### 5.1. Datasets and Experimental Setup

**Datasets.** We conduct most experiments on QM9S (Zou et al., 2023), an extension of QM9 (Ramakrishnan et al., 2014) with simulated multi-modal spectra (IR, Raman, and UV/Vis). QM9 contains about 134k small molecules with DFT-optimized 3D geometries. We use all three modalities as input and predict joint 2D–3D molecular structure. QM9S is preprocessed as in MolSpectra (Wang et al., 2025a).

**Evaluation metrics.** For structure elucidation, we report Acc@1, MCES, Morgan fingerprint similarities (TaniSimMG and CosSimMG), MACCS fingerprint similar-

*Table 3.* Ablation summary on QM9S. Unless otherwise specified, results are obtained with MAST *without* MCTS.

| Method | Acc@1↑ | MCES↓ | TaniSimMG↑ | CosSimMG↑ | TaniSimMA↑ | FraggleSim↑ | FGSim↑ | RMSD(Å)↓ | MapAcc↑ |
|---|---|---|---|---|---|---|---|---|---|
| *(A) Motif prior* | | | | | | | | | |
| MAST (w/o Motif) | 69.94% | 0.9783 | 0.8275 | 0.8737 | 0.9394 | 0.9601 | 0.9720 | 0.8136 | 90.34% |
| MAST (w/ Motif) | 79.12% | 0.6296 | 0.8834 | 0.9141 | 0.9596 | 0.9651 | 0.9806 | 0.8020 | 93.83% |
| *(B) Guidance and MCTS (MAST w/ Motif)* | | | | | | | | | |
| Guidance-only (fixed $\gamma$) | 79.32% | 0.6161 | 0.8864 | 0.9171 | 0.9618 | 0.9681 | 0.9817 | 0.7949 | 93.92% |
| Guidance-only (scheduled $\gamma(t)$) | 79.96% | 0.6287 | 0.8838 | 0.9153 | 0.9602 | 0.9678 | 0.9808 | 0.8011 | 93.70% |
| MCTS-guidance | **94.89%** | **0.1048** | **0.9774** | **0.9827** | **0.9908** | **0.9754** | **0.9935** | **0.7843** | **95.34%** |

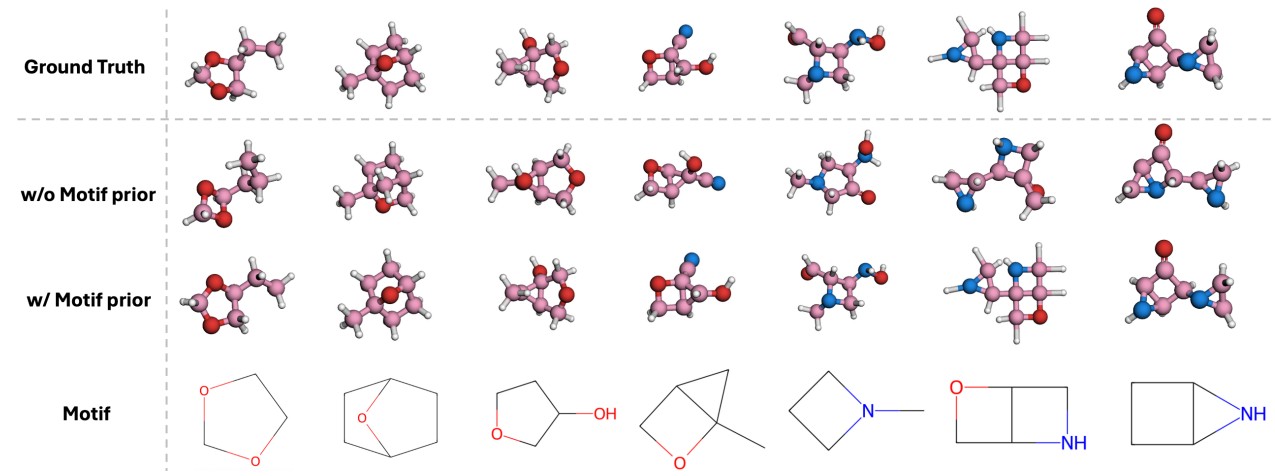

*Figure 3.* **Motif prior improves exact recovery.** Examples comparing ground-truth molecules with samples generated without or with the motif prior. The last row shows predicted motif cues that cannot be recovered unless they are incorporated as priors.

ity (TaniSimMA), fragment and functional-group similarity (FraggleSim and FGSim), and 3D geometry metrics (RMSD and MapAcc), following DiffSpectra (Wang et al., 2025b). Acc@1 is computed by converting the generated molecule object into canonical SMILES and checking exact match with the ground-truth. For RMSD, we allow mirror inversion because the spectra used are enantiomer-invariant. For generative quality, we report stability and distribution metrics including AtomStable, MolStable, V&U, V&U&N (Valid, Unique and Novel) (Huang et al., 2023b), FCD (Preuer et al., 2018), SNN, Frag, and Scaf (Polykovskiy et al., 2020). More details are provided in Appendix D.

**Baseline.** For structure elucidation, we compare against spectra-to-SMILES models, including IR-to-Structure (Alberts et al., 2024), Spectra2Structure (Kanakala et al., 2024), and LLM-based SpectraLLM (Su et al.), as well as diffusion-based generators such as DiffSpectra (Wang et al., 2025b). For molecular generation quality, we further report molecular diffusion generators including CDGS (Huang et al., 2023a) and JODO (Huang et al., 2023b).

**Implementation details.** All experiments are conducted on 8 NVIDIA RTX 3090 GPUs. For MCTS-guided tree search, we first follow MolSpectra (Wang et al., 2025a) to train molecule and spectra encoders for guidance and scoring. the UCB exploration coefficient $c_{\text{ucb}}$ is set to 1.0. For efficiency, we set the total number of denoising steps per

sample to at most 3000. We set the tree expansion width to 4 and the temperature for softmax backup $\tau = 0.5$. Additional implementation details are provided in Appendix C.

### 5.2. Benchmarking Molecular Structure Elucidation

**Structure elucidation.** We benchmark spectra conditioned structure elucidation on QM9S in Table 1. MAST demonstrates strong improvements over spectra-to-SMILES baselines and diffusion generators. Table 1 suggests two key conclusions. First, MAST contributes significantly more to structural resolution under spectral conditions than previous diffusion baseline methods. Without tree search, it improves Top-1 exact recovery (Acc@1) by 30.4 percentage points over the strongest baseline, and consistently achieves higher fingerprint- and fragment-level similarities. Second, tree search further amplifies performance with a frozen generator, reaching 94.89% Acc@1, 0.1048 MCES and higher similarities, while improving 3D fidelity with lower RMSD and higher MapAcc. These gains arise purely at test time, indicating that tree search can prioritize promising prefixes and avoid computation spent on low-consistency samples.

**Generative quality.** Table 2 summarizes sample stability and distributional fidelity on QM9S. MAST maintains near-perfect AtomStable and MolStable, indicating that improved conditioning and inference-time search do not compromise chemical validity. Tree-search further increases the fraction

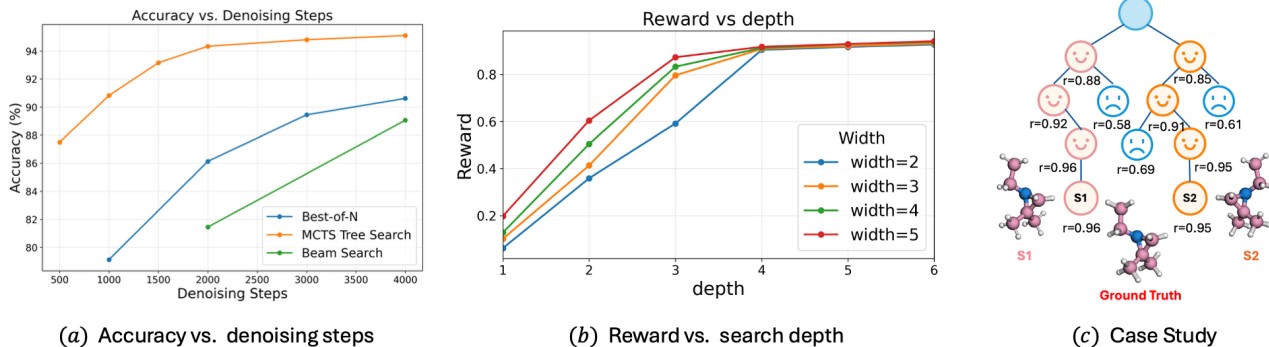

(a) Accuracy vs. denoising steps    (b) Reward vs. search depth    (c) Case Study

*Figure 4.* **MCTS-guided diffusion analysis.** (a) Exact recovery (%) under increasing denoising budgets. (b) Average spectra–structure reward versus search depth under different tree widths. (c) Visualization of a reward-guided search tree.

*Table 4.* Quality of motif evidence on QM9S.

| Setting | Motif-P↑ | Motif-R↑ | F1 (samples)↑ | F1 (macro)↑ |
|---|---|---|---|---|
| *(A) Spectra-to-motif predictor $E_{\mathrm{mot}}$* | | | | |
| $E_{\mathrm{mot}}$ (predictor) | 0.9563 | 0.9644 | 0.9529 | 0.9361 |
| *(B) Motif inclusion in generated molecules* | | | | |
| *Overall (all test cases)* | | | | |
| w/o Motif (gen) | 0.9595 | 0.9539 | 0.9503 | 0.9087 |
| w/ Motif (gen) | **0.9783** | **0.9699** | **0.9682** | **0.9430** |
| *SMILES-mismatch subset (hard cases)* | | | | |
| w/o Motif (gen) | 0.8600 | 0.8406 | 0.8280 | 0.7376 |
| w/ Motif (gen) | **0.8894** | **0.8462** | **0.8377** | **0.7667** |

of valid, unique, and novel molecules to $97.1\%$, compared with $90.5\%$ for DiffSpectra. Across distribution metrics, MAST consistently improves over prior diffusion baselines. For example, it achieves markedly lower FCD while improving nearest-neighbor and fragment/scaffold similarities, suggesting that the method preserves the underlying molecular manifold while selecting spectra-consistent samples.

### 5.3. Analysis: Motif Prior

We examine the effect of introducing an explicit motif prior as intermediate evidence for spectra-conditioned elucidation. Table 3 shows that removing the motif prior consistently degrades the performance: exact recovery drops and MCES increases, with consistent declines across structural similarity metrics. Qualitatively, Figure 3 indicates that, without the prior, samples may miss key motifs and drift toward incorrect structures; injecting the predicted motif cues steers generation toward spectra-consistent molecules. Some predicted motif cues are highlighted in the last row.

To understand why the motif prior helps, Table 4 evaluates motif evidence from reliability and utilization. First, the spectra-to-motif predictor $E_{\mathrm{mot}}$ achieves strong precision/recall and F1, indicating that multi-spectra contain learnable motif-level evidence. Notably, this provides more reliable motif-level signals than samples generated without the motif prior, suggesting that learning such intermediate evidence purely through end-to-end generation is non-trivial. Second, conditioning on the motif prior improves motif in-

clusion in generated molecules both on the full test set and the mismatch subset, demonstrating that the generator effectively exploits the motif prior to resolve hard ambiguities (e.g., overall macro-F1 rises from $0.9087$ to $0.9430$). Finally, a targeted analysis of cases corrected by adding the motif prior shows that $65\%$ of the corrections coincide with motif differences, spanning 100 of the 103 motif types, suggesting that the benefit from motif prior is broad.

### 5.4. Analysis: MCTS-guided Inference

Table 3 (Panel B) shows that guidance-only variants yield only marginal gains, suggesting that a single fixed or scheduled guidance scales is often insufficient. Figure 4 (a) further compares tree-search diffusion with Best-of-$N$ and Beam Search under increasing denoising budgets. Tree search consistently achieves higher exact recovery at the same step budget and improves faster as the budget increases. Best-of-$N$ wastes computation on redundant trajectories and prefixes that already deviate early, while Beam Search is less competitive under limited budgets since maintaining a beam requires multiple costly continuations per prefix.

We further profile the wall-clock runtime and memory usage of different inference strategies in Appendix G. The results show that MCTS-guided diffusion uses the inference budget more effectively than repeated-sampling baselines, rather than simply relying on substantially larger memory or longer runtime. Under comparable denoising-step and runtime budgets, MCTS-guided diffusion achieves a better accuracy–runtime trade-off than Best-of-N and Beam Search. For example, at around 2000 denoising steps, MCTS-guided diffusion reaches $94.34\%$ Acc@1 with $85.40$s per sample, while Best-of-N obtains $86.13\%$ Acc@1 with $79.68$s. Peak GPU memory remains similar across methods because they share the same denoising backbone. MCTS uses moderately more CPU memory to maintain the search tree and node statistics, but denoising remains the dominant cost.

Figure 4 (b) characterizes how spectra-structure reward varies with search depth and tree width, averaged over 512

randomly sampled test cases. Increasing the search depth yields consistent reward improvements, but the gains saturate after moderate depth. In contrast, increasing the tree width primarily benefits shallow depths by broadening early exploration and improving the chance of discovering high-reward prefixes, after which different widths converge as the search becomes dominated by a few promising subtrees. Figure 4 (c) visualizes a reward-guided tree on a representative example. The search concentrates visits on two high-reward candidates: one is almost identical to the ground-truth structure, while the other corresponds to a mirror-symmetric configuration that remains highly aligned with the spectra. This example highlights the non-uniqueness of spectra-conditioned elucidation and shows that our method naturally returns multiple high-reward, spectra-consistent solutions.

### 5.5. Robustness and Sensitivity Analyses

**Robustness to spectral perturbations.** Realistic spectra often contain measurement noise, baseline shifts, and other acquisition artifacts, which may distort peak intensities and weaken the spectra–structure correspondence. To evaluate whether MAST remains reliable beyond the clean QM9S setting, we perturb the input spectra with additive Gaussian noise and baseline drift. Full results are reported in Appendix E. MAST without MCTS degrades gracefully under both perturbation types. We further compare MAST with DiffSpectra under combined Gaussian noise and baseline drift. Across all tested perturbation settings, MAST consistently outperforms DiffSpectra by a large margin. These results verify motif-level evidence can be reliably extracted by the motif predictor even when the raw spectra are corrupted. These results indicate that motif-augmented conditioning provides robust intermediate chemical evidence under spectral shifts, which is an important source of the overall robustness of MAST.

**Sensitivity of reward and motif evidence.** We also analyze the sensitivity of MCTS-guided inference and motif conditioning in Appendix F. Softmax backup performs best among the tested backup strategies, suggesting that value propagation should emphasize high-reward rollouts while avoiding relying solely on the maximum reward. A moderate motif-consistency reward weight gives the best exact recovery, while too small or too large weights reduce performance. This indicates that motif consistency is most effective as a complementary constraint to the spectra-structure reward, rather than as the dominant objective. Moreover, MAST remains stable under mild perturbations of the spectra-structure reward scorer, showing that the search procedure does not require a perfectly calibrated reward model. In contrast, corrupting the predicted motif prior leads to clear performance drops, confirming that accurate motif evidence is important for resolving spectra-conditioned ambiguity. These analyses show that MAST benefits from both informative motif conditioning and reward-guided tree search, and that the final improvement is not caused by a single isolated component.

## 6. Conclusion

In this work, we present MAST, a motif-augmented diffusion framework with search tree for 3D molecular elucidation from spectra. Our experiments demonstrate that MAST substantially improves exact recovery and 3D reconstruction on QM9S. Additional robustness evaluations show that MAST remains effective under noisy spectral perturbations, while runtime–memory profiling confirms that MCTS-guided inference provides a favorable accuracy–efficiency trade-off compared with repeated sampling baselines. Future work can further scale MAST to broader chemical spaces, incorporate more spectral modalities, and evaluate the framework on more diverse datasets, especially real-world experimental spectra.

## Acknowledgements

This work is partially supported by the Robotic AI-Scientist Platform program of the Chinese Academy of Sciences.

## Impact Statement

This paper presents MAST, a motif-augmented diffusion framework with reward-guided tree search for spectroscopic molecular structure elucidation. By generating spectra-consistent candidate 2D/3D molecular structures, it may support higher-throughput chemical and materials characterization where spectra serve as indirect structural fingerprints. Potential benefits include assisting metabolite annotation, impurity/degradant characterization in drug development, and spectral interpretation in materials analysis. Moreover, The probabilistic motif prior provides interpretable mid-level evidence that may improve transparency and human-in-the-loop review.

Potential negative impacts include misuse and over-reliance: improved spectra-to-structure proposals could be combined with downstream synthesis or acquisition pipelines in sensitive contexts. In addition, spectra-to-structure mapping is inherently one-to-many, so the system may return chemically plausible but incorrect candidates especially under distribution shift. Since our experiments focus on simulated spectra, applying MAST to experimental data may require calibration, domain adaptation, and careful post-generation filtering and validation. We recommend using MAST as decision support by returning a ranked set of candidates, reporting uncertainty, and verifying results with orthogonal evidence. For sensitive deployments, appropriate access control and auditing should be considered.

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

**Roadmap.** This supplementary material complements the main text with the following components:

- **Model details** (Appendix A): formal specifications of the multi-spectra encoder, spectra-to-motif prior, conditioning mechanism, and joint 2D–3D denoiser.

- **Motif vocabulary construction** (Appendix B): extraction protocol, filtering criteria, and representative motifs in the vocabulary.

- **Implementation details** (Appendix C): training and inference configurations, including diffusion settings, encoder architectures, motif predictor training, and MCTS hyperparameters.

- **Evaluation metrics** (Appendix D): definitions and formulas for molecular generation quality, spectra-conditioned structure elucidation, and 3D geometry evaluation.

- **Robustness to spectral perturbations** (Appendix E): evaluations under additive Gaussian noise and baseline drift, including comparisons with DiffSpectra and analysis of motif-predictor robustness.

- **Additional sensitivity analyses** (Appendix F): studies on reward backup strategies, motif-consistency reward weights, reward-scorer perturbations, and motif-prior corruption.

- **Efficiency and resource profiling** (Appendix G): wall-clock runtime, GPU memory, and CPU memory comparisons under different denoising-step budgets.

## A. Model Details

A molecule is represented as a joint structure $G := (X, V, E)$, where $X \in \mathbb{R}^{N \times 3}$ are 3D coordinates, $V$ denotes node attributes (e.g., atom types and charges), and $E$ denotes edge/bond attributes (e.g., bond types). We denote the observed multi-modal spectra by $\mathcal{S} = \{s^{(k)}\}_{k=1}^{K}$, $s^{(k)} \in \mathbb{R}^{L_k}$.

### A.1. Multi-spectra Encoder

**Patching.** For each modality $k$, we split $s^{(k)} \in \mathbb{R}^{L_k}$ into a patch sequence $p^{(k)} \in \mathbb{R}^{N_k \times P_k}$ with patch length $P_k$ and stride $D_k$, where $N_k = \left\lfloor \frac{L_k - P_k}{D_k} \right\rfloor + 1$. Each patch is projected to $d$ dimensions by a learnable matrix $W_k \in \mathbb{R}^{P_k \times d}$ and added with positional embeddings $W_{\text{pos}}^{(k)} \in \mathbb{R}^{N_k \times d}$:

$$z^{(k)} = p^{(k)} W_k + W_{\text{pos}}^{(k)} \in \mathbb{R}^{N_k \times d}. \tag{19}$$

**Multi-spectrum Transformer.** We concatenate patch embeddings from all modalities: $Z_0 = \text{Concat}(z^{(1)}, \ldots, z^{(K)}) \in \mathbb{R}^{N_{\text{tot}} \times d}$, and apply a Transformer (Vaswani et al., 2017) encoder $f_{\text{spec}}$:

$$Z = f_{\text{spec}}(Z_0) \in \mathbb{R}^{N_{\text{tot}} \times d}. \tag{20}$$

We obtain a global spectrum embedding via pooling (e.g., mean-pooling):

$$h_{\text{spec}} = \text{Pool}(Z) \in \mathbb{R}^{d}. \tag{21}$$

### A.2. Spectra-to-Motif Prior

**Vocabulary and labels.** We define a vocabulary of $K_{\text{mot}}$ motifs $\mathcal{V} = \{v_1, \ldots, v_{K_{\text{mot}}}\}$. For a molecule $G$, we extract a multi-hot label $y(G) \in \{0, 1\}^{K_{\text{mot}}}$ with

$$y_k(G) = \mathbb{I}[v_k \subseteq G]. \tag{22}$$

**Predictor.** A multi-label predictor $E_{\text{mot}}$ maps spectra to logits $\ell_{\text{mot}} \in \mathbb{R}^{K_{\text{mot}}}$ and probabilities $p_{\text{mot}} \in (0, 1)^{K_{\text{mot}}}$:

$$\ell_{\text{mot}} = E_{\text{mot}}(s), \qquad p_{\text{mot}} = \sigma(\ell_{\text{mot}}). \tag{23}$$

We implement $E_{\text{mot}}$ as a hierarchical Transformer that first captures local spectral patterns within each modality and then models the global and cross-modal dependencies via self-attention over concatenated patches. The training loss is (weighted) binary cross-entropy:

$$\mathcal{L}_{\text{mot}} = -\sum_{k=1}^{K_{\text{mot}}} \Big( w_k \, y_k \log p_k + (1 - y_k) \log(1 - p_k) \Big). \tag{24}$$

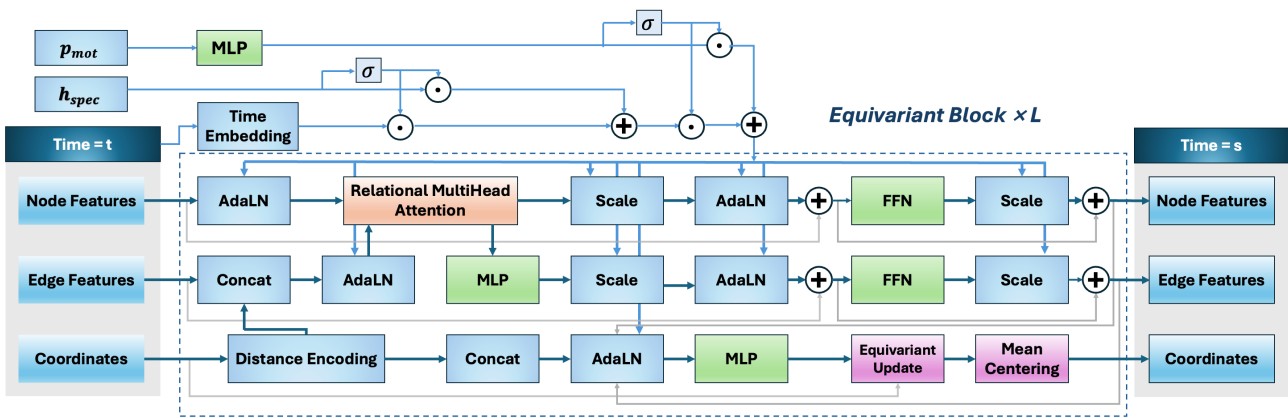

*Figure 5.* **Denoising network architecture.** Detailed architecture of the SE(3)-equivariant denoiser used in MAST. The model conditions on the multi-spectra embedding $h_{\text{spec}}$, the motif prior $p_{\text{mot}}$, and the diffusion timestep $t$, and iteratively updates node features, edge features, and 3D coordinates through $L$ stacked equivariant blocks.

### A.3. Conditioning Embedding

Diffusion is conditioned on spectral evidence $c := (h_{\text{spec}}, p_{\text{mot}})$ together with diffusion time $t$. The time $t$ is embedded as $\tau(t) \in \mathbb{R}^{d_h}$.

**Two-stage gate fusion.** Conditioning is fused by two successive vector-valued gates (Fig. 5). First, the spectral embedding and time embedding are fused by a gate-weighted sum:

$$g_s(s) = \sigma\Big(\psi_{g,s}\big( h_{\text{spec}} \big)\Big) \in (0,1)^{d_h}, \tag{25}$$

$$c_{st}(t,s) = g_s(s) \odot h_{\text{spec}} + \big(1 - g_s(s)\big) \odot \tau(t) \in \mathbb{R}^{d_h}, \tag{26}$$

where $\psi_{g,s}(\cdot)$ is a MLP, $\sigma(\cdot)$ is the sigmoid, and $\odot$ denotes element-wise multiplication.

Next, the motif probabilities are mapped to a dense embedding

$$z_{\text{mot}} = \psi_{\text{mot}}(p_{\text{mot}}) \in \mathbb{R}^{d_h}, \tag{27}$$

and fused with $c_{st}(t,s)$ by another gate-weighted sum:

$$g_c(s) = \sigma\Big(\psi_{g,c}\big( z_{\text{mot}} \big)\Big) \in (0,1)^{d_h}, \tag{28}$$

$$c_{\text{fuse}}(t,s) = g_c(s) \odot c_{st}(t,s) + \big(1 - g_c(s)\big) \odot z_{\text{mot}} \in \mathbb{R}^{d_h}. \tag{29}$$

where $\psi_{g,c}(\cdot)$ is a MLP, $\sigma(\cdot)$ is also the sigmoid, and $\odot$ also denotes element-wise multiplication.

**Injection into denoiser blocks.** The final conditioning vector $c_{\text{fuse}}(t,s)$ is used to modulate all denoiser blocks. Concretely, for each equivariant block, $c_{\text{fuse}}(t,s)$ is mapped to the adaptive modulation parameters (e.g., AdaLN shift/scale and the multiplicative Scale coefficients) that control the node-, edge-, and coordinate-update pathways.

### A.4. Joint 2D–3D Denoiser with Conditional Equivariant Blocks

Let the noisy state at time $t$ be $G_t = (X_t, V_t, E_t)$. The denoiser is a conditional mapping

$$\hat{G}_0 = D_\theta(G_t; C(t,s)) = (\hat{X}_0, \hat{V}_0, \hat{E}_0), \tag{30}$$

implemented by stacking $L$ blocks that update node stream, edge stream, and coordinate stream jointly.

**Adaptive normalization and scaling.** We apply AdaLN(Peebles & Xie, 2023) and conditional scaling with conditioning vector $C$:

$$\text{AdaLN}(h,C) = \big(1 + f_{\text{sc}}(C)\big) \odot \text{LN}(h) + f_{\text{bs}}(C), \tag{31}$$

$$\text{Scale}(h,C) = g(C) \odot h, \tag{32}$$

where $f_{\text{sc}}, f_{\text{bs}}, g$ are small MLPs and $\odot$ denotes elementwise product.

**Relational attention.** Let $H^{(l)} \in \mathbb{R}^{N \times d_h}$ be node features at layer $l$ and $A_{ij}^{(l)} \in \mathbb{R}^{d_a}$ be edge features. We augment edges with invariant geometry (e.g., squared distances):

$$\bar{A}_{ij}^{(l)} = \left[ A_{ij}^{(l)} \parallel \|X_i^{(l)} - X_j^{(l)}\|_2^2 \parallel \Phi(\|X_i^{(l)} - X_j^{(l)}\|_2^2) \right], \tag{33}$$

where $\Phi(\cdot)$ is a fixed or learnable basis expansion (e.g., Gaussian basis). A relational multi-head attention produces an aggregated message $M^{(l)} \in \mathbb{R}^{N \times d_h}$:

$$M^{(l)} = \mathrm{R-MHA}\Big( \mathrm{AdaLN}(H^{(l)}, C), \ \mathrm{AdaLN}(\bar{A}^{(l)}, C) \Big). \tag{34}$$

**Node and edge updates.** We update nodes via residual + FFN:

$$H^{(l+1)} = \mathrm{Scale}(M^{(l)}, C) + H^{(l)}, \tag{35}$$

$$H^{(l+1)} = \mathrm{Scale}\Big( \mathrm{FFN}\big(\mathrm{AdaLN}(H^{(l+1)}, C)\big), C \Big) + H^{(l+1)}. \tag{36}$$

Edge features are updated by fusing node messages and applying AdaLN/FFN:

$$\hat{A}_{ij}^{(l)} = \big( M_i^{(l)} + M_j^{(l)} \big) W_1, \tag{37}$$

$$A_{ij}^{(l+1)} = \mathrm{Scale}(\hat{A}_{ij}^{(l)}, C) + A_{ij}^{(l)}, \tag{38}$$

$$A_{ij}^{(l+1)} = \mathrm{Scale}\Big( \mathrm{FFN}\big(\mathrm{AdaLN}(A_{ij}^{(l+1)}, C)\big), C \Big) + A_{ij}^{(l+1)}. \tag{39}$$

**Coordinate update (equivariant field).** Coordinates are updated by aggregating relative displacement vectors:

$$e_{ij}^{(l)} = \mathrm{AdaLN}\Big( W_2 \big[ H_i^{(l+1)} \| H_j^{(l+1)} \| A_{ij}^{(l+1)} \| \|X_i^{(l)} - X_j^{(l)}\|_2^2 \big], \ C \Big), \tag{40}$$

$$X_i^{(l+1)} = X_i^{(l)} + \sum_{j \neq i} \gamma^{(l)} \frac{X_i^{(l)} - X_j^{(l)}}{\|X_i^{(l)} - X_j^{(l)}\|_2^2} \ \tanh\big( \mathrm{FFN}(e_{ij}^{(l)}) \big), \tag{41}$$

where $\gamma^{(l)}$ is a learnable scalar. Finally we mean-center coordinates to remove global translation.

### A.5. Diffusion Training Objective

We use data prediction and regress the clean structure from noisy inputs. For a random $t \in [0,1]$, we sample noise and obtain $G_t$ via the VP marginal (component-wise). The denoiser predicts $\hat{G}_0$, and we minimize a weighted reconstruction loss:

$$\mathcal{L}_{\text{diff}} = \mathbb{E}\Big[ \lambda_X \|X - \hat{X}_0\|_2^2 + \lambda_V \|V - \hat{V}_0\|_2^2 + \lambda_E \|E - \hat{E}_0\|_2^2 \Big]. \tag{42}$$

## B. Motif Vocabulary Construction

We construct a motif vocabulary as an explicit meso-scale prior between spectra and molecular graphs. Following the knowledge-base extraction protocol in K-MSE (Zhuang et al., 2025), we extract two complementary families of motifs—*ring* and *chain* patterns—and represent each motif as a SMILES string. Concretely, for ring motifs, we enumerate rings, merge rings that share at least two atoms, and include one-hop attachments to preserve immediate chemical context; for chain motifs, we collect two-hop neighborhoods centered at non-ring carbon atoms and filter out candidates containing any ring atoms to keep them purely acyclic. All extracted motifs are canonicalized by RDKit and counted on QM9S; we retain motifs that appear in more than $1\%$ of molecules, yielding a compact vocabulary of $K_{\text{mot}} = 103$ motifs.

This vocabulary spans diverse intermediate-level chemistry (local topology, ring membership, and functional-group-bearing fragments such as ethers/amines/nitriles/carbonyls), providing a structured prior that is more informative than atom types yet far more compact than full-graph templates; representative motifs are shown in Table 5.

Table 5. **Representative motifs** in our motif vocabulary, grouped by coarse chemical semantics.

| Category | Representative motifs (SMILES) | Chemical cue |
|---|---|---|
| Alkyl / branching | CCC, CC(C)C, CC(C)(C)C | hydrophobic backbone, steric bulk |
| Unsaturation (alkene/alkyne) | C=CC, CC=CC, C#CC | $\pi$-bond patterns |
| Alcohols / polyols | CCO, CC(O)O, CC(O)CO | H-bond donors/acceptors, polarity |
| Ethers / acetals | COC, CCOC, COC(C)OC | H-bond acceptors, flexible linkage |
| Amines (primary/tertiary/cyclic) | CCCN, CN(C)C, CN1CCCC1 | basicity; HBD/HBA sites |
| Carbonyls (aldehyde/ketone) | CCC=O, CC(C)C=O, CC(C)=O | polar carbonyl center |
| Carboxylic acids / esters | CC(=O)O, CCC(=O)O, COC(C)=O | acidity / ester functionality |
| Nitriles | CCC#N, CCCC#N, CC(C)C#N | strong H-bond acceptor |
| Aromatic / heteroaromatic | C1=CCCC1, C1=CCOC1, C1=COC=N1 | conjugated ring electronics |
| Small rings / saturated heterocycles | C1CC1, C1CCOC1, C1CCNC1 | rigidity; ring/heteroatom context |

Table 6. **Functional groups used for FGSim**.

| Category | Groups included in FGSim |
|---|---|
| C skeleton | alkane_sp3_CH; alkane; alkene; alkyne |
| O-containing | aliphatic_alcohol; ether; epoxide; aldehyde; ketone; ester; carbonyl_any |
| N-containing | amine_any; primary_amine; secondary_amine; tertiary_amine; amide_like; nitrile |
| F-containing | fluoroalkane; fluoro_any |

# C. Implementation Details

All experiments are conducted on $8 \times$ NVIDIA RTX 3090 GPUs. For spectral conditioning, we employ a Transformer-style encoder over three modalities (UV, IR, Raman) with depth 4, width 256, 8 heads, and output dimension 128. Spectra are patched with lengths $(P_{\text{uv}}, P_{\text{ir}}, P_{\text{ram}}) = (32, 64, 64)$ and strides $(16, 32, 32)$. We pretrain the spectral encoder with masked autoencoding (MAE) using a mask ratio of 0.7, and then fine-tune it jointly with the denoiser. We incorporate an explicit spectra-to-motif prior with a vocabulary of $K_{\text{mot}} = 103$ motifs. The motif predictor is trained separately using binary cross-entropy with uniform weights $w_k = 1.0$, and is kept frozen during diffusion training.

We use a VP-SDE with a cosine schedule(Song et al., 2020), setting $\beta_0 = 0.1$ and $\beta_1 = 20.0$, and sample with $T = 1000$ denoising steps. Our denoiser include an SE(3)-equivariant Transformer with hidden width $d_h = 256$, $L = 8$ equivariant blocks and 16 attention heads. Training uses AdamW (Loshchilov & Hutter, 2017) with learning rate $2 \times 10^{-4}$, $\beta_1 = 0.9$, $\epsilon = 10^{-8}$ and dropout rate is 0.1. The weights of node, edge, and coordinate losses are $\lambda_{\text{V}} = 1.0$, $\lambda_{\text{E}} = 0.25$, and $\lambda_{\text{X}} = 0.1$.

For MCTS-guided inference, we follow the MolSpectra (Wang et al., 2025a) pipeline and train molecule and spectrum encoders to compute spectrum–structure consistency rewards. We discretize reverse diffusion into 50 linear checkpoints and perform Monte Carlo Tree Search (MCTS) over checkpointed segments. Unless otherwise stated, we run MCTS with UCB exploration coefficient $c = 1.0$, and expand up to 4 children per node. Each node uses a lightweight rollout of 10 denoising steps per segment to estimate value, and we cap the total number of denoising steps per sample at 3000 for efficiency. We adopt a `softmax` backup strategy with temperature $\tau = 0.5$ to aggregate subtree returns. For guidance, we use a small scale set $\gamma \in \{0.0, 0.1, 0.15, 0.2, 0.25, 0.3, 0.4\}$. To ensure chemical validity during search, we set the node reward to zero when the 2D structure is invalid; if the 2D structure is valid but the 3D geometry is invalid, we downweight the reward by a factor of 0.5.

# D. Evaluation Metrics

We evaluate our method from two complementary perspectives: (i) *molecular generation quality*, measuring chemical validity/stability and distributional fidelity of generated molecules; and (ii) *spectra-conditioned structure elucidation*, measuring exact recovery and graded structural/geometry similarity. Unless otherwise stated, we follow the metric implementations used in DiffSpectra (Wang et al., 2025b).

### D.1. Molecular Generation Quality

**Validity, uniqueness, and novelty.** We generate $n_{\text{gen}}$ molecules and decode them into chemically-typed molecular graphs. A molecule is *valid* if it passes standard chemistry checks. We canonicalize each valid molecule and count duplicates as

*Table 7.* Robustness of MAST (w/o MCTS) under additive Gaussian noise and baseline drift.

| Perturbation setting | Acc@1↑ | TaniSimMG↑ | CosSimMG↑ | TaniSimMA↑ | FGSim↑ | MapAcc↑ |
|---|---|---|---|---|---|---|
| Clean | 79.12% | 0.8834 | 0.9141 | 0.9596 | 0.9806 | 93.83% |
| Gaussian noise (SNR=50 dB) | 77.73% | 0.8593 | 0.8915 | 0.9441 | 0.9707 | 78.81% |
| Gaussian noise (SNR=40 dB) | 76.76% | 0.8556 | 0.8902 | 0.9481 | 0.9758 | 77.82% |
| Gaussian noise (SNR=30 dB) | 75.98% | 0.8493 | 0.8850 | 0.9456 | 0.9750 | 77.03% |
| Baseline drift=3 | 77.15% | 0.8565 | 0.8896 | 0.9434 | 0.9708 | 78.42% |
| Baseline drift=4 | 76.17% | 0.8492 | 0.8847 | 0.9449 | 0.9761 | 77.23% |
| Baseline drift=5 | 74.61% | 0.8373 | 0.8747 | 0.9358 | 0.9682 | 75.64% |

*Table 8.* Robustness of the motif predictor under noisy and shifted spectra.

| Setting | micro-F1↑ | macro-F1↑ |
|---|---|---|
| Clean | 0.9726 | 0.9484 |
| SNR=20 dB, drift=5 | 0.9661 | 0.9401 |
| SNR=30 dB, drift=5 | 0.9679 | 0.9417 |
| SNR=40 dB, drift=5 | 0.9683 | 0.9434 |

non-unique. A molecule is *novel* if its canonical form does not appear in the training set. Let $n_{\text{v\&u}}$ be the number of *valid and unique* molecules and $n_{\text{v\&u\&n}}$ the number of *valid, unique, and novel* molecules. We report

$$\text{V\&U} = \frac{n_{\text{v\&u}}}{n_{\text{gen}}}, \qquad \text{V\&U\&N} = \frac{n_{\text{v\&u\&n}}}{n_{\text{gen}}}. \tag{43}$$

These ratios jointly penalize invalid decoding and mode collapse, while V&U&N additionally reflects generalization.

**AtomStable and MolStable.** We report AtomStable and MolStable to quantify stability at atom- and molecule-level, respectively, using the standard valence/coordination criteria adopted in prior 3D molecular generators and DiffSpectra (Huang et al., 2023b; Wang et al., 2025b).

**Distribution-based metrics.** We assess distributional fidelity between generated samples and a reference set using standard metrics. **FCD** (Fréchet ChemNet Distance) measures Fréchet distance between ChemNet feature distributions (Preuer et al., 2018). **SNN** (Similarity to Nearest Neighbor), **Frag** (BRICS fragment statistics), and **Scaf** (Bemis–Murcko scaffold statistics) follow the MOSES benchmark definitions (Polykovskiy et al., 2020). Higher SNN/Frag/Scaf and lower FCD indicate better distributional alignment.

## D.2. Spectra-conditioned Structure Elucidation

**Acc@K (exact recovery).** For each spectrum, we obtain $K$ decoded candidates $\{\hat{G}_i\}_{i=1}^K$ and a ground-truth molecule $G$. We compute Acc@K as the fraction of test cases where at least one candidate matches the ground truth exactly:

$$\text{Acc@}K = \mathbb{E}\left[\mathbb{I}\big(\exists i \in \{1, \ldots, K\} \text{ s.t. } \hat{G}_i = G\big)\right]. \tag{44}$$

In practice, we convert each decoded molecule object to a *canonical SMILES* using RDKit and check exact match against the ground-truth canonical SMILES.

**Connectivity and fingerprint similarities.** We report MCES as a graded measure of graph discrepancy, together with Morgan fingerprint similarities (TaniSimMG and CosSimMG), MACCS fingerprint similarity (TaniSimMA), FraggleSim, and FGSim, following DiffSpectra (Wang et al., 2025b). These metrics quantify partial structural overlap when exact match is not achieved. For FGSim, we extract a set of chemically meaningful functional/motif groups using SMARTS-based substructure matching and compute set-level overlap (Table 6).

**3D geometry metrics.** We evaluate 3D consistency using RMSD and MapAcc (atom-mapping accuracy), following DiffSpectra (Wang et al., 2025b). RMSD is computed after establishing an atom correspondence under atom-type constraints and rigid alignment and we allow mirror inversion because the spectra used are enantiomer-invariant. MapAcc measures the fraction of correctly matched atoms under the established correspondence.

*Table 9.* Comparison with DiffSpectra under combined Gaussian noise and baseline drift.

| Perturbation setting | Method | Acc@1↑ | TaniSimMG↑ | CosSimMG↑ | TaniSimMA↑ | FGSim↑ |
|---|---|---|---|---|---|---|
| SNR=20 dB, drift=5 | DiffSpectra | 0.20% | 0.0285 | 0.0476 | 0.0665 | 0.0887 |
| SNR=20 dB, drift=5 | MAST (w/o MCTS) | 74.41% | 0.8322 | 0.8698 | 0.9301 | 0.9692 |
| SNR=30 dB, drift=3 | DiffSpectra | 7.68% | 0.2576 | 0.3582 | 0.4413 | 0.5428 |
| SNR=30 dB, drift=3 | MAST (w/o MCTS) | 78.02% | 0.8554 | 0.8890 | 0.9452 | 0.9746 |
| SNR=30 dB, drift=4 | DiffSpectra | 10.43% | 0.2574 | 0.3510 | 0.4460 | 0.5224 |
| SNR=30 dB, drift=4 | MAST (w/o MCTS) | 76.04% | 0.8422 | 0.8798 | 0.9432 | 0.9782 |
| SNR=30 dB, drift=5 | DiffSpectra | 9.38% | 0.2790 | 0.3795 | 0.4643 | 0.5488 |
| SNR=30 dB, drift=5 | MAST (w/o MCTS) | 73.83% | 0.8303 | 0.8687 | 0.9322 | 0.9741 |
| SNR=40 dB, drift=5 | DiffSpectra | 51.95% | 0.8316 | 0.8776 | 0.9376 | 0.9264 |
| SNR=40 dB, drift=5 | MAST (w/o MCTS) | 74.41% | 0.8347 | 0.8732 | 0.9340 | 0.9711 |

*Table 10.* Effect of different reward backup strategies in MCTS.

| Backup strategy | Acc@1↑ | TaniSimMG↑ | CosSimMG↑ | TaniSimMA↑ | FraggleSim↑ | FGSim↑ | RMSD(Å)↓ |
|---|---|---|---|---|---|---|---|
| Softmax | 95.70% | 0.9895 | 0.9921 | 0.9970 | 0.9787 | 0.9968 | 0.7768 |
| Max | 94.92% | 0.9855 | 0.9894 | 0.9961 | 0.9714 | 0.9959 | 0.7646 |
| Mean | 94.53% | 0.9809 | 0.9836 | 0.9947 | 0.9707 | 0.9947 | 0.7757 |

## E. Robustness to Spectral Perturbations

The main experiments in the paper are conducted on clean simulated spectra from QM9S. However, spectra obtained in practical scenarios may contain various acquisition artifacts, such as random measurement noise, imperfect background correction, and baseline shifts. These perturbations can distort peak intensities and global spectral shapes, making the inverse mapping from spectra to molecular structures more ambiguous. Therefore, we further evaluate whether MAST can preserve spectra–structure consistency when the input spectra are corrupted.

We consider two common perturbation types. First, we add Gaussian noise to the input spectra, where the perturbation strength is controlled by the signal-to-noise ratio (SNR). Lower SNR corresponds to stronger noise. Second, we add baseline drift to mimic systematic shifts in the spectral baseline. Compared with random noise, baseline drift changes the global spectral profile and can be particularly harmful for models that rely heavily on global spectral embeddings. All robustness experiments in this section are performed without retraining the model, so they directly test the robustness of the learned spectra-conditioned generator.

Table 7 reports the robustness of MAST without MCTS under individual perturbations. The results show that MAST degrades gracefully as the perturbation strength increases. For Gaussian noise, Acc@1 decreases from 79.12% on clean spectra to 75.98% under $SNR = 30$ dB, while structural similarity metrics such as TaniSimMA and FGSim remain high. For baseline drift, Acc@1 decreases to 74.61% at drift = 5, but the generated structures still preserve strong fingerprint and functional-group similarity. This indicates that even when exact recovery becomes harder, MAST tends to generate molecules that remain structurally close to the ground truth.

We further compare MAST with DiffSpectra under combined Gaussian noise and baseline drift in Table 9. This setting is more challenging because the spectra are affected by both local random fluctuations and global baseline shifts. MAST consistently outperforms DiffSpectra across all perturbation settings. For example, under $SNR = 30$ dB and drift = 5, MAST achieves 73.83% Acc@1, while DiffSpectra obtains only 9.38%. Even under the severe $SNR = 20$ dB and drift = 5 setting, MAST maintains 74.41% Acc@1, whereas DiffSpectra nearly collapses. These results suggest that motif-augmented conditioning provides more robust intermediate evidence than relying only on global spectral representations.

Finally, Table 8 evaluates whether the motif predictor itself remains reliable under noisy and shifted spectra. The micro-F1 and macro-F1 scores drop only slightly compared with the clean setting, indicating that the spectra-to-motif mapping is relatively stable even when the raw spectra are corrupted. This robustness of the motif predictor is an important reason for the robustness of MAST. Since the predicted motif prior provides intermediate local chemical evidence during generation, a stable motif predictor allows the denoising model to rely on chemically meaningful constraints rather than only on perturbed global spectral features. As a result, MAST can still preserve spectra–structure consistency under noisy and shifted inputs.

## F. Additional Sensitivity Analyses

We conduct additional sensitivity analyses to better understand which design choices are important for MCTS-guided diffusion. These experiments focus on four aspects: the reward backup strategy, the motif-consistency reward weight,

Table 11. Sensitivity to perturbations in the spectra–structure reward model.

| Reward perturbation setting | Acc@1↑ | TaniSimMG↑ | CosSimMG↑ | TaniSimMA↑ | FraggleSim↑ | FGSim↑ | RMSD(Å)↓ |
|---|---|---|---|---|---|---|---|
| Clean scorer | 95.70% | 0.9895 | 0.9921 | 0.9970 | 0.9787 | 0.9968 | 0.7768 |
| Gaussian noise, std=0.02 | 94.92% | 0.9848 | 0.9894 | 0.9672 | 0.9823 | 0.9952 | 0.7774 |
| Gaussian noise, std=0.05 | 93.36% | 0.9791 | 0.9858 | 0.9651 | 0.9823 | 0.9932 | 0.7893 |
| Gaussian noise, std=0.10 | 88.28% | 0.9434 | 0.9586 | 0.9531 | 0.9761 | 0.9926 | 0.7931 |

Table 12. Effect of the motif-consistency reward weight.

| Motif reward weight | Acc@1↑ | TaniSimMG↑ | CosSimMG↑ | TaniSimMA↑ | FraggleSim↑ | FGSim↑ | RMSD(Å)↓ |
|---|---|---|---|---|---|---|---|
| 0.00 | 94.14% | 0.9810 | 0.9862 | 0.9952 | 0.9822 | 0.9945 | 0.7888 |
| 0.05 | 95.31% | 0.9872 | 0.9903 | 0.9968 | 0.9782 | 0.9952 | 0.7798 |
| 0.10 | 95.70% | 0.9895 | 0.9921 | 0.9970 | 0.9787 | 0.9968 | 0.7768 |
| 0.20 | 94.53% | 0.9855 | 0.9895 | 0.9959 | 0.9787 | 0.9972 | 0.7820 |

the robustness of the spectra–structure reward scorer, and the quality of the motif prior. Unless otherwise specified, the sensitivity experiments are conducted on 1024 samples with the same trained generator.

**Reward backup strategy.** During MCTS, rollout rewards need to be propagated back to previously visited nodes. The backup rule determines how the value of a node is estimated from its observed rollout rewards. Table 10 compares softmax, max, and mean backup. Softmax backup achieves the best Acc@1. This suggests that the search benefits from emphasizing high-reward rollouts, but using only the maximum reward may be too sensitive to noisy or lucky evaluations. Mean backup is more conservative, but it can dilute strong signals from promising branches. Therefore, softmax backup provides a useful compromise between robustness and exploitation.

**Perturbing the reward scorer.** Table 11 evaluates the sensitivity of MCTS to perturbations in the spectra–structure reward model. Mild perturbations only slightly reduce performance, indicating that the search procedure does not require a perfectly calibrated scorer. However, stronger perturbations lead to a larger drop in Acc@1. This is expected because the reward scorer determines which denoising prefixes are expanded more often. If the scorer becomes too noisy, MCTS may allocate computation to suboptimal branches.

**Motif-consistency reward weight.** Table 12 studies the weight of the motif-consistency reward in the MCTS evaluation function. When the weight is 0, the search relies only on the spectra–structure reward. Adding a moderate motif reward improves Acc@1, with the best result obtained at weight 0.10. This confirms that motif consistency provides complementary guidance during tree search. However, increasing the weight to 0.20 reduces exact recovery. One possible explanation is that an overly strong motif constraint may overemphasize local motif agreement and weaken the role of the global spectra–structure reward, which is necessary for exact molecular recovery.

**Perturbing the motif prior.** Table 13 evaluates the effect of corrupting the predicted motif prior before generation. Small Gaussian perturbations cause moderate performance degradation, while random bit flips and fully random motif priors lead to much larger drops. For example, when the flip rate is increased to $r = 0.3$, Acc@1 drops to $51.37\%$, and using a random motif prior further reduces Acc@1 to $29.49\%$. These results confirm that the motif prior is not merely an auxiliary input, but an important source of interpretable chemical evidence. Wrong motif prior can substantially reduce exact recovery.

# G. Efficiency and Resource Profiling

Inference-time scaling improves performance by spending additional computation at test time. However, for diffusion models, such computation can be expensive because each candidate sample requires a sequential denoising trajectory. Therefore, it is important to compare not only accuracy under different denoising-step budgets, but also wall-clock runtime and memory usage. In this section, we profile Best-of-N, Beam Search, and MCTS-guided diffusion under different budgets.

Table 14 reports the accuracy–runtime–memory trade-off. Best-of-N increases accuracy by drawing more independent full-chain samples, but this strategy repeatedly explores redundant trajectories and cannot reuse partially successful denoising prefixes. Beam Search maintains multiple partial candidates, but it still needs to continue several candidates at each stage, which makes it less efficient under limited budgets. In contrast, MCTS-guided diffusion expands the denoising tree adaptively. It allocates more computation to prefixes that receive higher spectra–structure rewards and prunes less promising

*Table 13.* Effect of perturbing the motif prior before generation.

| Motif prior setting | Acc@1↑ | TaniSimMG↑ | CosSimMG↑ | TaniSimMA↑ | FraggleSim↑ | FGSim↑ | MapAcc↑ |
|---|---|---|---|---|---|---|---|
| MAST (w/o MCTS) | 79.12% | 0.8834 | 0.9141 | 0.9596 | 0.9651 | 0.9806 | 93.83% |
| + Motif prior noise, $\sigma = 0.5$ | 78.52% | 0.8618 | 0.8921 | 0.9434 | 0.8251 | 0.9728 | 79.60% |
| + Motif prior noise, $\sigma = 1.0$ | 78.32% | 0.8607 | 0.8918 | 0.9441 | 0.8277 | 0.9756 | 79.41% |
| + Motif prior noise, $\sigma = 2.0$ | 77.93% | 0.8590 | 0.8905 | 0.9432 | 0.8273 | 0.9716 | 79.01% |
| + Motif prior flip, $r = 0.1$ | 72.27% | 0.8186 | 0.8581 | 0.9188 | 0.8055 | 0.9471 | 73.47% |
| + Motif prior flip, $r = 0.3$ | 51.37% | 0.6580 | 0.7269 | 0.8157 | 0.7535 | 0.8882 | 52.48% |
| + Random motif prior | 29.49% | 0.4585 | 0.5493 | 0.6629 | 0.6557 | 0.7767 | 30.10% |

*Table 14.* Accuracy–runtime–memory comparison under different denoising-step budgets.

| Method | Denoising steps | Acc@1 (%)↑ | Avg. runtime / sample (s) | Peak GPU memory (MiB) | Peak RAM (MiB) |
|---|---|---|---|---|---|
| Best-of-N | 1000 | 79.12 | 40.43 | 1584.1 | 5364.5 |
| Best-of-N | 2000 | 86.13 | 79.68 | 1584.3 | 5364.8 |
| Best-of-N | 3000 | 89.45 | 118.94 | 1583.8 | 5363.7 |
| Best-of-N | 4000 | 90.62 | 158.27 | 1584.2 | 5365.1 |
| Beam Search | 2000 | 81.45 | 81.74 | 1644.0 | 5423.0 |
| Beam Search | 4000 | 89.06 | 162.03 | 1646.2 | 5424.8 |
| MCTS-guided diffusion | 1000 | 90.82 | 47.44 | 1622.5 | 6025.0 |
| MCTS-guided diffusion | 2000 | 94.34 | 85.40 | 1626.5 | 6027.4 |
| MCTS-guided diffusion | 3000 | 94.81 | 124.63 | 1625.0 | 6030.7 |
| MCTS-guided diffusion | 4000 | 95.11 | 163.03 | 1623.0 | 6038.9 |

branches early. As a result, it achieves better accuracy under comparable runtime.

For example, at around 3000 denoising steps, MCTS-guided diffusion reaches $94.81\%$ Acc@1 with $124.63$s per sample, while Best-of-N reaches $89.45\%$ Acc@1 with $118.94$s. Similarly, at around 4000 denoising steps, MCTS-guided diffusion reaches $95.11\%$ Acc@1, outperforming both Best-of-N and Beam Search. The runtime of MCTS-guided diffusion is slightly higher at the same nominal step budget because it involves tree operations and reward evaluations, but the accuracy gain is much larger than this overhead.

The peak GPU memory is similar across all methods because the denoising backbone dominates GPU memory consumption and is shared by all inference strategies. The main additional memory cost of MCTS appears in CPU RAM, since the search tree stores checkpoint states, visit counts, values, and other node statistics. This explains why MCTS-guided diffusion uses moderately higher RAM than Best-of-N and Beam Search.

Overall, the profiling results show that MCTS-guided diffusion uses the inference budget more effectively. Rather than relying on substantially larger GPU memory or much longer runtime, MCTS improves performance by adaptively allocating computation to high-reward denoising prefixes. Consequently, under comparable denoising-step and runtime budgets, MCTS achieves consistently higher exact recovery than Best-of-N and Beam Search.

