# OpenReview forum: "MAST: Motif-Augmented Diffusion with Search Tree for Spectroscopic Molecular Structure Elucidation"
_ICML.cc/2026/Conference — ICML 2026 regular_

### Official Review · Reviewer_GpQQ · 2026-03-10

**Soundness:** 3
**Presentation:** 3
**Significance:** 3
**Originality:** 4
**Overall Recommendation:** 5
**Confidence:** 3

**Summary:**

This paper proposes MAST, a framework for reconstructing molecular 2D and 3D structures from multimodal spectra. The key idea is to introduce explicit motif priors into a diffusion model, and to formulate inference as a discrete decision process guided by Monte Carlo Tree Search. On QM9S, MAST reportedly achieves up to 94.89% Top-1 reconstruction accuracy, substantially outperforming prior generative baselines.

**Compliance With Llm Reviewing Policy:**

Affirmed.

**Final Justification:**

This paper proposes MAST, combining explicit chemical motif priors with a diffusion model and MCTS-style search for multimodal spectra-conditioned 2D–3D molecular structure elucidation. Overall, the work is technically solid, reasonably clear, and potentially impactful for spectra-to-structure generation, particularly due to the interpretable motif integration and budget-aware inference.

Strengths include: (1) strong empirical performance with informative ablations and budgeted inference comparisons (MCTS vs Best-of-N/beam), (2) improved interpretability via motif intermediates, and (3) large gains in exact recovery while maintaining chemical validity/stability.

Key weaknesses remain: (1) inference is computationally heavy (long diffusion trajectories plus tree search), which may raise latency concerns in high-throughput settings. (2) Also, while results support the chosen design, the rebuttal does not fully prove that diffusion plus the specific MCTS pruning is uniquely necessary versus other simpler refinement/acceleration strategies.

Rebuttal impact on my concerns: it largely addressed my main questions and reinforced my prior positive assessment. For Q1, the authors added an explicit comparison to an earlier AR Transformer baseline using the same motif and spectral inputs, showing very low Acc@1 (1.17%) versus MAST, supporting the need for the diffusion formulation in their setting. For Q2, they clarified the mathematical consistency of discretizing diffusion into segment-level actions by packaging the full sequence of Gaussian draws within a segment; transitions are deterministic conditioned on the sampled action while the overall process remains stochastic. This resolves my earlier concern about conceptual inconsistency and frames the method as finite-budget best-path discovery rather than expectation approximation.

Overall, despite remaining compute/latency limitations, the method and insights (domain priors + diffusion + budgeted search) are valuable to the community, so I recommend **Accept**.

**Key Questions For Authors:**

1. Task complexity and model choice: If the motif predictor already resolves most motifs and global structure, is a diffusion model necessary for coordinate refinement? Have you compared against simpler alternatives such as direct regression or rule-based assembly after motif identification, and what is the compute–accuracy trade-off?
2. Discretizing continuous diffusion into tree search actions: Does mapping continuous denoising randomness into discrete actions introduce information loss or early commitment that could trap the search in suboptimal regions? Given that diffusion trajectories may converge to similar optima across different noise paths, how do you justify the need to explore many trajectories, and what evidence supports the correctness of the pruning strategy?
3. Explaining poor baseline performance: Why do the recent baselines fail so severely under your evaluation? Is it due to inability to recover 3D coordinates or mismatches in structure matching? Please provide a detailed failure analysis.

**Limitations:**

yes

**Strengths And Weaknesses:**

Strengths:
1. Comprehensive ablations and comparisons: The paper compares Monte Carlo Tree Search, Best-of-N sampling, and beam search under different denoising budgets, which is informative for high-throughput settings and supports the compute allocation advantages of Monte Carlo Tree Search.
2. Strong interpretability: Using 103 explicit chemical motifs as intermediate representations enables a spectra to motifs to structure pipeline that aligns with domain reasoning and improves transparency relative to latent-only approaches.
3. Large performance gains: The method improves Acc@1 from around 60% in prior work to nearly 95% while maintaining chemical validity.

Weaknesses and Limitations:
1. High computational cost: Achieving peak performance requires up to 3000 denoising steps per sample, and the overall inference cost remains substantial.
2. Latency concerns: The combination of long diffusion trajectories, tree search, and multiple encoders may limit applicability in real-time or industrial high-throughput scenarios.
3. Unclear baseline gap: Recent baselines in the comparison reportedly achieve near-zero accuracy, while MAST exceeds 90%, raising concerns about evaluation protocol differences and requiring clarification.

---

> ### Author Rebuttal · Authors · 2026-03-30
>
> Thank you for your detailed and constructive feedback. We address your concerns below.
>
> > **W1: High computational cost.**
>
> **Reply:** We agree that diffusion-based generation is computationally heavier than autoregressive SMILES prediction. However, in our setting, spectra-to-SMILES baselines perform much worse than diffusion-based methods. We therefore view the added cost as a reasonable trade-off for the large gain in exact recovery and structural fidelity. At the same time, our use of **MCTS** is precisely motivated by this issue: rather than simply increasing compute, it improves performance under the same inference budget by allocating computation more effectively across denoising trajectories.
>
> > **W2: Latency concerns.**
>
> **Reply:** We acknowledge the latency concern. Importantly, our added **accuracy--runtime** comparison shows that, under **comparable wall-clock runtime**, our method still achieves higher accuracy. In practice, throughput can also be improved by batching across samples and parallelizing branch evaluation, which can partially mitigate the latency overhead in higher-throughput settings.
>
> > **W3: Unclear baseline gap.**
>
> **Reply:** We believe the gap mainly comes from the fact that spectra-to-SMILES methods cast the task as **1D symbolic translation**, whereas molecular elucidation from spectra is fundamentally a **structured 2D/3D recovery problem**.
>
> Direct SMILES generation is especially fragile in this setting. Spectral evidence is typically ambiguous, local, and distributed across the molecule, but autoregressive string decoding requires committing to a single linearized representation step by step. As a result, small early errors can propagate and cause incorrect connectivity or ring structure, even when the generated molecule remains chemically related to the target. By contrast, our diffusion model operates directly on the joint molecular graph and 3D geometry, which is better matched to the structure of the problem.
>
> We will clarify this reasoning in the revised version.
>
> > **Q1: If the motif predictor already resolves most motifs and global structure, is a diffusion model still necessary?**
>
> **Reply:** Yes. The motif predictor provides a strong **intermediate prior**, but it is still far from a complete molecular structure. It only indicates which motifs are likely present, while the model still needs to determine how these motifs are connected, assign the exact atom/bond configuration, and recover a valid 3D geometry. In other words, the gap from motif-level evidence to a full 2D/3D molecule remains substantial.
>
> As discussed in **W3**, directly generating a 1D symbolic representation is a difficult formulation for this task. In earlier stages of this project, we also explored **Transformer-based autoregressive prediction** as a simpler alternative, but its performance was not competitive. This is why we ultimately adopted a diffusion-based generator.
>
> > **Q2: Why is tree-search discretization reasonable?**
>
> **Reply:** **First, it does not introduce essential information loss.** We do not replace diffusion with a coarse discrete model; we only make branching decisions at selected checkpoints, while the within-segment evolution still follows the original stochastic diffusion process.
>
> **Second, it helps avoid premature commitment rather than causing it.** Different branches use different random noise realizations at the initial expansion, and later expansions also introduce different random draws, so the search still explores diverse trajectories rather than collapsing to a single path. Moreover, MCTS explicitly balances **exploration** and **exploitation**, which further reduces the chance of getting trapped in suboptimal regions too early.
>
> **Third, exploring multiple trajectories is necessary because the spectra-conditioned posterior can be multi-modal.** Diffusion stochasticity naturally enables different plausible solutions rather than redundant copies, and **Figure 4(c)** shows a representative case where two highly explored paths correspond to two different yet spectra-consistent structures.
>
> **Finally, pruning is introduced to reduce the cost of this exploration.** By downweighting low-value branches early, MCTS allocates more computation to promising trajectories. As supported by **Figure 4(a)** and **Figure 4(c)**, this allows the search to achieve higher recovery under the same budget by reducing computation wasted on low-reward paths.
>
> > **Q3: Why do recent baselines fail so severely? Provide a failure analysis.**
>
> **Reply:** As discussed in **W3**, the main issue is insufficiently precise structural recovery under the spectra-to-SMILES formulation. In practice, these baselines often generate chemically related but structurally incorrect molecules.
>
> One representative example is that for the target SMILES `O=CNc1cocn1`, **SpectraLLM** generates `O=Cc1c[nH]cnc1=O`, which is already incorrect at the molecular structure level.

---

> > ### Author Rebuttal · Reviewer_GpQQ · 2026-04-02
> >
> > Thank you for your response, but issues remain.
> >
> > 1) **R to Q1.** Your explanation is a reasonable motivation, but it still does not provide evidence that diffusion is necessary over simpler assembly/regression/refinement baselines. Even without new experiments, please briefly summarize the earlier AR-Transformer baseline with main performance numbers or cite closely related “simpler prior” approaches you compared against.
> >
> > 2) **R to Q2.** The claim that “within-segment evolution still follows the original stochastic diffusion process” appears conceptually inconsistent with the manuscript’s formulation: Sec. 4.2 explicitly states that once an action fixes the noise realization (packaged into the action), the segment transition becomes deterministic (Eq. 10). My core statistical concern remains: the high-dimensional continuous Gaussian noise space is discretized into a very small set of candidate actions (e.g., width \(w=4\)), which seems insufficient to approximate the relevant expectation, inducing high variance and early commitment to a few random seeds. Please provide a formal mathematical description and justification for this discretization and its implications.
> >
> > Overall, I feel the current responses are more conceptual defenses than evidence-backed clarifications.

---

> > > ### Author Response · Authors · 2026-04-03
> > >
> > > Thanks for the follow-up questions. We provide additional replies as follows.
> > >
> > > > **Q1: Is diffusion still necessary beyond a motif predictor and a simpler AR baseline?**
> > >
> > > **Reply:** To provide evidence-backed explanation, we compare MAST with our earlier **AR baseline**, which takes the same **motif prior** and **spectral information** as input and directly predicts a **SMILES** sequence.
> > >
> > > |Method|Acc@1↑|TaniSimMG↑|CosSimMG↑|TaniSimMA↑|FraggleSim↑|FGSim↑|
> > > |---|---:|---:|---:|---:|---:|---:|
> > > |Earlier AR baseline|1.17%|0.1497|0.2593|0.5645|0.6464|0.8362|
> > > |MAST (w/o MCTS)|79.12%|0.8834|0.9141|0.9596|0.9651|0.9806|
> > >
> > > Even with the same motif prior and spectral input, the AR baseline is far less accurate than MAST on all evaluation metrics. Results of other **spectra-to-SMILES** methods under the same setting have already been reported in **Table 1**, and metric definitions are provided in **Appendix D**. Overall, these results show that the diffusion-based formulation provides a clear advantage over simpler autoregressive alternatives.
> > >
> > > > **Q2: Why is tree discretization reasonable?**
> > >
> > > **Reply:** First, there is no inconsistency between our wording and Eq. (10). In our formulation, each segment-level action is defined as $a_\ell=(\xi_\ell,\gamma_\ell)$, where $\gamma_\ell$ is the segment guidance strength and $\xi_\ell$ denotes the **full collection of Gaussian noise draws within that segment**. If a segment contains 20 reverse steps, then one sampled action corresponds to one realization of those 20 stochastic updates, i.e., $\xi_\ell=(\epsilon_{\ell,1},\dots,\epsilon_{\ell,20})$. This action is generated by exactly the same reverse-diffusion sampler as the original process; the only difference is that we package the 20-step realization into one segment-level action. Therefore, once $a_\ell$ is generated, the transition
> > > $$
> > > h_{\ell+1}=F_\theta(h_\ell;a_\ell,c)
> > > $$
> > > is deterministic w.r.t. the checkpoint state and action, exactly as stated in Eq. (10). The overall process remains stochastic because $a_\ell$ itself is sampled. Thus, the segment tree does not change the original stochastic diffusion process.
> > >
> > > Second, our method is not designed to approximate the full high-dimensional Gaussian expectation with a small sampling set. Instead, it is a **finite-budget best-path discovery** procedure. Under a denoising-step budget $B$, let each full trajectory cost $T$ steps and each segment cost $s$ steps. Then a repeated full-sampling baseline such as Best-of-$N$ can explore at most $N=\lfloor B/T\rfloor$ full trajectories, whereas our tree search can already expand $w$ candidate actions at the first checkpoint as long as $ws\le B$. Let $p_1$ be the probability that a random $1^{st}$ segment action lands on a **good prefix**, then the probability of covering at least one good prefix at the first segment is
> > > $$
> > > P_{\text{tree},1}=1-(1-p_1)^w,
> > > $$
> > > while for Best-of-$N$ it is
> > > $$
> > > P_{\text{BoN},1}=1-(1-p_1)^N.
> > > $$
> > > In principle, because one segment is much cheaper than one full trajectory, $ w $ can be set much larger than $ N $ under the same denoising-step budget. In our setting, the total budget is 3000 denoising steps, a full trajectory uses 1000 steps, and the tree width is $w=4$. Therefore Best-of-$N$ can only take $N=3$, while our method achieves
> > > $$
> > > P_{\text{tree},1}=1-(1-p_1)^4 > 1-(1-p_1)^3=P_{\text{BoN},1}.
> > > $$
> > > So under the same budget, tree search has strictly better **shallow-level prefix coverage**. For example, if $p_1=0.7$, then
> > > $$
> > > 1-(1-0.7)^4=0.9919.
> > > $$
> > > This shows that when the proposal distribution assigns moderate mass to good prefixes, even a small width can provide high shallow-level coverage.
> > >
> > > Third, besides broader prefix coverage, our method is **more efficient in accumulation of promising prefixes**. Let $p_\ell$ denote the conditional probability that a random action at level $\ell$ extends a good prefix in a promising direction (i.e. into another good prefix). Then the probability that a correct branch survives through the first $L$ segments is approximately
> > > $$
> > > P_{\text{tree},\le L}\approx \prod_{\ell=1}^{L}\big(1-(1-p_\ell)^w\big).
> > > $$
> > > Unlike Best-of-$N$, our checkpointed search evaluates prefixes and reallocates later computation toward those with higher estimated accumulation value through UCB selection and reward backpropagation. As a result, computation is increasingly focused on prefixes with higher accumulated probability than unguided full-trajectory sampling.
> > >
> > > Finally, this is also supported empirically. Across 5 runs with different random seeds, the exact recovery ranges only from **94.53%** to **94.91%**, showing very small variation. Moreover, Fig. 4(b) shows that although increasing tree width improves performance, the marginal gain quickly diminishes once the width exceeds 4, and the curves for different widths become nearly consistent as the depth increases. This suggests that our method does not suffer from severe early lock-in in practice, and that the proposed discretization is stable.

---

### Official Review · Reviewer_fnr8 · 2026-03-11

**Soundness:** 3
**Presentation:** 3
**Significance:** 3
**Originality:** 3
**Overall Recommendation:** 4
**Confidence:** 4

**Summary:**

This paper proposes MAST, a framework for spectra-conditioned joint 2D/3D molecular structure elucidation. The method augments a conditional SE(3)-equivariant diffusion model with two main components: (1) a separate spectra-to-motif predictor that outputs a probabilistic motif prior, which is injected into the denoising network alongside a global spectral embedding, and (2) an inference-time checkpointed MCTS procedure that searches diffusion trajectories using a learned spectra-structure alignment reward and a motif-consistency reward. On QM9S, the authors report that MAST improves Acc@1 from 60.68% for their reproduced DiffSpectra baseline to 79.12% without MCTS, and further to 94.89% with MCTS, while also improving several similarity and 3D reconstruction metrics.

**Compliance With Llm Reviewing Policy:**

Affirmed.

**Final Justification:**

The authors have provided strong experimental results and analysis in the rebuttal, which can clarify my previous concerns. Thus I will increase my rating.

**Key Questions For Authors:**

1. The paper reports a reproduced DiffSpectra Acc@1 of 60.68%, whereas the DiffSpectra abstract reports 40.76% top-1 and 99.49% top-10. Can the authors explain the exact differences in data split, preprocessing, candidate generation, ranking, and evaluation protocol?

2. Can the authors provide any evidence beyond QM9S simulated spectra, such as evaluation on experimental spectra, synthetic-to-experimental transfer, or at least controlled noise / calibration / domain-shift experiments?

3. How sensitive are the results to motif vocabulary construction choices, such as the 1% frequency threshold and K=103 motifs? Relatedly, what happens with oracle motifs, corrupted motifs, or end-to-end rather than frozen motif training?

**Limitations:**

Yes

**Strengths And Weaknesses:**

## Strengths

MAST adapts the motif prior, which is a sensible way to externalize spectra-to-structure ambiguity into an interpretable intermediate representation. The paper also validates that this component matters: removing the motif prior causes a large drop in Acc@1 (79.12% to 69.94%), and Table 4 further shows that the motif predictor is nontrivial and improves motif-consistent generation.

I think another strength is the inference-time search design. Reformulating diffusion sampling as a checkpointed search tree and combining alignment guidance with MCTS is an interesting test-time scaling perspective for generative modeling. More broadly, the paper integrates motif-level conditioning and diffusion-tree search into a joint 2D/3D spectra-conditioned generation framework in a technically coherent way. While the individual ingredients are not entirely new, their combination is meaningful and practically relevant.

## Weakness

My main concerns are about the scope of the empirical claims and the significance of the results. Almost all evidence is on QM9S, which uses simulated IR/Raman/UV-Vis spectra for relatively small molecules. The paper acknowledges that experimental deployment would require calibration and domain adaptation, but does not provide results on experimental spectra or a convincing synthetic-to-experimental transfer study. This substantially weakens the practical significance of the claims.

The fairness of some comparisons is also not fully clear. Table 1 compares the tri-modal MAST system against several single-modality spectra-to-SMILES baselines, so DiffSpectra appears to be the strongest fair comparator. In addition, the headline 94.89% result relies on MCTS with up to 3000 denoising steps and a separate MolSpectra-style reward model, so it should be interpreted as a model-plus-search-budget result rather than purely a generator result. Figure 4 helps, but the main text should more clearly separate model quality from extra inference-time compute, ideally with matched-budget or matched-wall-clock comparisons.

It remains somewhat unclear how much of the gain comes from tree search itself, from the auxiliary scorer, from reward design, or simply from additional compute. Stronger ablations on reward terms, oracle/noisy motif priors, motif vocabulary design, and frozen vs joint motif training would strengthen the paper.

---

> ### Author Rebuttal · Authors · 2026-03-29
>
> Thank you for your detailed and constructive feedback. We address your concerns below.
> > **W1 & Q2: Limited evidence beyond QM9S.**
>
> **Reply:**  Experimental spectra paired with reliable 3D conformations are currently scarce, so we use QM9S as the main benchmark and add two robustness evaluations.
>
> First, we perturb the input spectra with **Gaussian noise** and **baseline drift** to mimic realistic measurement errors. Results are shown in **MAST under noisy spectra** (https://anonymous.4open.science/r/2026-B258/MAST_Noise_Spectra.png).
>
> We also compare against **DiffSpectra** under combined perturbations, with results shown in **DiffSpectra vs. MAST under noisy spectra** (https://anonymous.4open.science/r/2026-B258/MAST_Diff_Noise.png).
>
> Second, we collect 25 **real experimental spectra** from the **NIST WebBook** for molecules, with results shown in **MAST on real spectra** (https://anonymous.4open.science/r/2026-B258/Real_data.png).
>
> Overall, these results show that our model is robust and performs far better than DiffSpectra on both corrupted and real spectra. With **MCTS**, **Acc@1** further improves to **78.12%** under **SNR=30 dB + drift=5** and **61.76%** on the real subset.
>
> This robustness is largely supported by **motif predictor**, whose performance drops only slightly under perturbations. See **motif predictor robustness** (https://anonymous.4open.science/r/2026-B258/Motif_noise.png).
> > **W2.1: Fairness of the baseline comparison.**
>
> **Reply:** Among spectra-to-SMILES methods, the strongest reported baseline is **SpectraLLM**, which reports joint three-modality results but still performs far below our method, e.g., **Tanimoto** (**0.3355** vs. **0.8834**); see **baseline comparison** (https://anonymous.4open.science/r/2026-B258/baseline.png). We will include this result as one of the baselines in the revised version.
> > **W2.2: The 94.89\% result is a model-plus-search-budget result.**
>
> **Reply:** We agree that the **94.89\%** result should be interpreted as a **model-plus-search-budget** result. Howerver, **MAST (w/o MCTS)** already substantially outperforms **DiffSpectra**, so the gain from **60.68\% \(\rightarrow\) 79.12\%** reflects the **generator itself**.
>
>  We additionally measure **wall-clock runtime** and report **accuracy--runtime curves** for Best-of-N, Beam Search, and MCTS. Under **comparable runtime**, our method still achieves higher accuracy. See **accuracy--runtime** (https://anonymous.4open.science/r/2026-B258/Acc-time.png).
>
> > **W3: Attribution of gains is unclear.**
>
> **Reply:** We agree that a finer-grained attribution is valuable.
>
> First, the gain from **motif conditioning** is already shown in **Table 3(A)**, where adding the motif prior improves Acc@1 from **69.94\%** to **79.12\%**.
>
> To further analyze the remaining gains, we add three ablations:
> - **backup strategy** (https://anonymous.4open.science/r/2026-B258/backup.png)
> - **motif reward weight** (https://anonymous.4open.science/r/2026-B258/weight.png)
> - **reward scorer perturbation** (https://anonymous.4open.science/r/2026-B258/scorer.png).
>
> These results show that
> - Search quality depends on the **backup strategy**, though all variants remain competitive.
> - A **moderate motif-consistency reward weight** works best.
> - The full system benefits from a high-quality **spectra-structure scorer**, while remaining relatively stable under mild perturbation.
>
> For **additional compute**, we provide the corresponding results in **W2.2**. For the motif-related factors, we provide a more detailed discussion in **Q3**.
> > **Q1: Why does reproduced DiffSpectra Acc@1 differ from the abstract?**
>
> **Reply:** The main difference is in the **evaluation protocol**. As stated in our paper, we **allow mirror inversion because the spectra are enantiomer-invariant**. Therefore, mirror-related 3D structures are both counted as correct. **Figure 4(c)** is an representative example.
> > **Q3: How sensitive are the results to motif priors, vocabulary, and training choices?**
>
> **Reply:** We agree that motif vocabulary design and training strategy are important. A complete study of **different vocabularies** or **training choices** would require retraining both the **motif predictor** and the **diffusion model**, which is beyond the rebuttal period.
>
> Our current design adopts a **1\% frequency threshold** as a trade-off between richer structural priors and reliable prediction. We keep the **motif predictor frozen** to preserve a stable and interpretable intermediate representation. The results in **W1** show that our predictor remains stable under noisy spectral.
>
> We also perform **corrupted-motif** experiments using Gaussian noise, random bit flips, and fully random motif priors. See **motif prior corruption** (https://anonymous.4open.science/r/2026-B258/motif_prior_corruption.png).
>
> These results show that motif-prior quality matters: heavy corruption causes large performance drops, while moderate perturbation leads to more gradual degradation.

---

> > ### Author Rebuttal · Reviewer_fnr8 · 2026-04-06
> >
> > I think the authors fully resolved my questions and concerns. Although using noise-augmented data as a substitute for experimental spectra presents certain limitations, this approach is necessitated by the nature of the experimental spectral dataset itself. Furthermore, the authors have provided an ablation study analyzing the various modules. Therefore, I consider this to be an excellent method that merits acceptance.

---

> > > ### Author Response · Authors · 2026-04-07
> > >
> > > We sincerely thank the reviewer for the very positive follow-up and for raising the rating. We truly appreciate the reviewer’s support and will incorporate the additional results and clarifications into the final version of the manuscript.

---

### Official Review · Reviewer_rdUm · 2026-03-12

**Soundness:** 2
**Presentation:** 2
**Significance:** 2
**Originality:** 2
**Overall Recommendation:** 2
**Confidence:** 4

**Summary:**

This paper proposes MAST, a diffusion framework for reconstructing 3D structures of molecules from multi-modal spectra. The two core contributions are: (1) a motif predictor that externalizes implicit local chemical information from spectra into an explicit motif probability vector, which is combined with a global spectral embedding to condition the diffusion denoising process, and (2) a checkpointed reverse diffusion tree over which MCTS is performed, guided by a spectra-structure alignment reward to enable test-time compute scaling. I appreciate the motivation of this work, but the experimental validation insufficient with potential data leakage risks, and the ablation results are confusing.

**Compliance With Llm Reviewing Policy:**

Affirmed.

**Final Justification:**

My core concern is that validation only on QM9S is insufficient to demonstrate the practical utility and generalizability of the proposed method. This is because the molecular spectra in QM9S are computationally simulated rather than experimentally measured, and the molecules in QM9S contain only CHONF atom types with fewer than 9 heavy atoms per molecule, a rather narrow slice of the broader chemical space.

To address my concern, the authors supplemented their rebuttal with results on NIST data. However, the molecules selected are still those overlapping with QM9S. While the new results do suggest that the method can generalize, to some extent, from computed to experimental spectra, the evaluation remains confined to QM9-like molecules. My concern is therefore only partially addressed.

**Key Questions For Authors:**

1. The original MolSpectra encoder was trained on the full QM9S dataset. Was the test split used for the diffusion model also applied when training your MolSpectra, training your motif predictor, or constructing motif vocabulary? Please clarify whether your method is affected by data leakage.
2. MAST(w/o Motif) achieves an Acc@1 of 69.94%, while the reproduced DiffSpectra achieves only 60.68%. What is the reason of this gap?

**Limitations:**

I encourage the authors to carefully examine and explicitly address/clarify potential data leakage risks, and to validate the method on more datasets and real experimental spectra to better demonstrate generalizability and practical applicability.

**Strengths And Weaknesses:**

**Strengths**

1. The motivation for incorporating motif priors into diffusion-conditioned generation is well-founded and clearly articulated.
2. Applying MCTS to diffusion inference as a form of test-time scaling represents a methodologically interesting contribution.
3. The ablation study covers both core components, the motif prior and MCTS, providing some insight into their individual contributions.

**Weaknesses**

1. Data leakage risk. The guidance and scoring in MCTS rely on the MolSpectra encoder, which was trained on the full QM9S dataset, including the test molecules used in this paper's experiments, meaning MolSpectra's training incorporated both the spectra and structures of the test set. Since the paper does not clarify whether MolSpectra was retrained on a modified dataset excluding the test split, there is a potential data leakage risk. Furthermore, it is not made clear whether the motif vocabulary construction and motif predictor training also used the full QM9S dataset including test molecules. If data leakage is present, the reported experimental results cannot be considered reliable.
2. Confusing ablation results. In the ablation study, MAST(w/o Motif) removes both the motif prior and MCTS sampling, and should therefore be equivalent to the baseline model DiffSpectra. Yet its Acc@1 is approximately 9% higher than the reproduced DiffSpectra. The source of this gap is not explained in the paper.
3. Overly idealized experimental setting. All experiments are conducted on simulated spectra, with no validation on real experimental spectra subject to noise, instrument variation, or measurement conditions. It is therefore difficult to assess the practical applicability of the proposed method.
4. Limited dataset coverage. Evaluation is conducted solely on QM9 (molecules with at most 9 heavy atoms), making it difficult to draw conclusions about the generalizability of the method.
5. Guidance provides negligible benefit. Table 3 (B) shows that adding guidance alone yields only marginal improvement.
6. The comparison in Figure 4(a) uses total denoising steps as the computational budget, which may not fully capture the actual inference cost of MCTS, as additional overhead from rollout evaluation, guidance backpropagation, and reward computation is not accounted for. A wall-clock time comparison against Best-of-N and Beam Search would be helpful to provide a more complete picture of the efficiency trade-offs.
7. Limited expressiveness of the motif label design. The multi-hot binary label cannot encode the count of motif occurrences, leading to information loss for molecules containing repeated instances of the same motif, which could produce misleading priors during generation. This limitation is relatively minor for the small molecules in QM9 (at most 9 heavy atoms), but would become increasingly problematic when scaling to medium- or large-sized molecules.

---

> ### Author Rebuttal · Authors · 2026-03-29
>
> Thank you for your detailed and constructive feedback. We address the raised weaknesses and questions below.
> > **W1 & Q1: Potential data leakage.**
>
> **Reply:** There is **no data leakage** in our work. The **MolSpectra encoder, motif predictor, and diffusion model use the same train/test split**, and the **motif vocabulary is constructed from the training split only**. We will state this more clearly in the revision.
> > **W2 & Q2: Unexplained gap between MAST (w/o Motif) and DiffSpectra.**
>
> **Reply:** `MAST (w/o Motif)` is **not intended to reproduce DiffSpectra**. As detailed in the Appendix, our backbone differs in how spectral information is injected into the denoising network. DiffSpectra uses an **MLP-based conditioning module**, while our model uses a more expressive **gated conditioning mechanism**. We will clarify this in the revision.
> > **W3: No validation on real experimental spectra.**
>
> **Reply:** To assess robustness beyond clean simulated spectra, we conduct two additional analyses.
>
> **First**, we perturb the input spectra with **Gaussian noise** and **baseline drift**. See **MAST under Gaussian noise and baseline drift** (https://anonymous.4open.science/r/2026-B258/MAST_Noise_Spectra.png).
>
> We also compare against **DiffSpectra** under combined perturbations. See **DiffSpectra vs. MAST under noisy spectra** (https://anonymous.4open.science/r/2026-B258/MAST_Diff_Noise.png).
>
> **Second**, we add **25 real experimental spectra** from the **NIST WebBook** for molecules in our test set. See **performance on the real spectra** (https://anonymous.4open.science/r/2026-B258/Real_data.png).
>
> Overall, these results show that our model is robust: it remains stable under noisy perturbations and performs far better than DiffSpectra on both corrupted and real spectra.
>
> This robustness is largely supported by the **motif predictor**, whose performance drops only slightly under perturbations. See **motif predictor robustness under noisy spectra** (https://anonymous.4open.science/r/2026-B258/Motif_noise.png).
>
> With **MCTS**, **Acc@1** further improves to **78.12%** under **SNR=30 dB + drift=5** and **61.76%** on the real subset.
> > **W4: Limited evaluation dataset coverage.**
>
> **Reply:** At present, there is no other public benchmark with paired **spectra + molecular structures + reliable 3D conformations**. Still, as shown in **W3**, our additional noisy-spectrum and real-spectrum experiments provide preliminary evidence beyond the clean in-domain setting. Together, these results indicate that our model and method are robust to realistic spectral perturbations and show promising generalization beyond the original QM9S benchmark.
> > **W5: Guidance brings little gain.**
>
> **Reply:** We believe this is mainly for two reasons. First, guidance is differentiable only w.r.t. **continuous coordinates**, so it mainly refines geometry, while many failures come from **atom-type errors**. Second, a fixed guidance strategy is suboptimal in conditional diffusion: the best guidance strength can vary across denoising stages. This also explains why **guidance + MCTS** works better, since MCTS can adaptively choose more suitable guidance strengths.
>
> > **W6: Inference cost is not fully captured.**
>
> **Reply:** In our implementation, the reported denoising-step budget already includes the cost of fast rollouts. We additionally measure **wall-clock runtime per sample** and report **accuracy--runtime curves** for Best-of-N, Beam Search, and MCTS-guided diffusion. See **accuracy--runtime** (https://anonymous.4open.science/r/2026-B258/Acc-time.png). The results show that, under comparable runtime, our method still achieves higher accuracy than Best-of-N and Beam Search.
>
> Our profiling also shows that denoising dominates the total cost, while reward evaluation and backpropagation are relatively minor. For example, under the **3000-step** MCTS setting, the **average number of simulations is only 48.2**.
> > **W7: Motif labels ignore counts.**
>
> **Reply:** The binary multi-hot motif prior is a deliberate design. Predicting **motif counts** would be more expressive but also much harder, and likely less reliable. In our setting, **prior reliability is more important** than extra detail.
>
> This is supported by an additional experiment: randomly flipping **30%** of motif bits drops **Acc@1** from **79.12%** to **51.37%**, showing that prior quality is crucial.
>
> Moreover, the motif prior is only an **auxiliary condition** rather than a complete specification of the final molecule. Its role is to provide coarse structural evidence. The diffusion model still needs to infer these finer details from the spectra, so the binary motif prior does not force an incorrect final structure even when count information is missing.
>
> Although the binary representation does not encode counts, our results show that motif presence/absence already provides a strong and useful prior. We will clarify this rationale and discuss count-aware representations as future work.

---

> > ### Author Rebuttal · Reviewer_rdUm · 2026-04-04
> >
> > I thank the authors for their rebuttal, and my concern about data leakage has been addressed. However, I still have some unresolved concerns.
> >
> > Regarding the validation on real experimental spectra, I think only 25 spectra samples cannot comprehensively reflect the generalizability and robustness of the proposed method. I checked the prior work Vib2Mol [a], which has already used more than 10,000 NIST experimental spectra for validation and also evaluated on other experimental spectra datasets such as SDBS. Therefore, I believe the lack of experimental spectra validation remains a weakness of this paper, which affects the evaluation of its practical usefulness. Additionally, since it appears that there are no 3D structures in the NIST database, how the proposed model is adapted to elucidate molecular structures should also be clarified.
> >
> > Regarding the ablation of `MAST (w/o Motif)`, thank you for pointing out the discrepancy in gated conditioning. Given the obvious effects of the gating mechanism, it would be better to report what gating coefficients are learned in this ablation and in the main experiment.
> >
> > In summary, I will keep my score until the paper further addresses these issues and reaches an acceptable level for publication.
> >
> > [a] Xinyu Lu, et al. Vib2Mol: From Vibrational Spectra to Molecular Structures—A Versatile Deep Learning Model

---

> > > ### Author Response · Authors · 2026-04-05
> > >
> > > > **Q1: More experiments on real experimental spectra are needed.**
> > >
> > > **Reply:** Thank you for this constructive comment and for pointing us to **Vib2Mol**. We agree that validation on real experimental spectra is important for assessing practical robustness.
> > >
> > > We would like to clarify the experimental setting in **Vib2Mol**. Although Vib2Mol collected experimental spectra from **NIST** and **SDBS**, the reported **test** sets contain only **602 samples from NIST** and **140 samples from SDBS**, while the remaining spectra are mainly used for **fine-tuning/training**. However, these two datasets do **not provide molecular conformation information**, so we cannot fine-tune our model on experimental spectra in the same way, because our framework requires paired **experimental spectrum-3D conformation** data.
> > >
> > > To address this issue as fairly as possible, we matched the public molecules shared between our **QM9S** test set and the molecules collected from **NIST/SDBS**, and obtained **258 paired samples** containing both **experimental spectra** and corresponding **molecular conformations**. We then evaluated both **MAST** and **DiffSpectra**.
> > >
> > > |Method|Acc@1↑|TaniSimMG↑|CosSimMG↑|TaniSimMA↑|FGSim↑|RMSD(Å)↓|MapAcc↑|
> > > |---|---:|---:|---:|---:|---:|---:|---:|
> > > |DiffSpectra|2.33|0.3205|0.4711|0.4909|0.7296|0.8798|0.4145|
> > > |MAST|44.19|0.5715|0.6363|0.6330|0.7832|0.7329|0.6342|
> > > |MAST(w/ MCTS)|60.47|0.7523|0.7949|0.7718|0.8560|0.6864|0.7684|
> > >
> > > As shown above, **MAST consistently outperforms DiffSpectra on experimental spectra**, and **MCTS** further improves the results. These results provide stronger evidence of robustness on real experimental spectra. We agree that with more paired **experimental spectra + 3D conformations** data used for **training or fine-tuning**, the performance of MAST can be further improved.
> > >
> > > It is also worth noting that **Vib2Mol** fine-tunes on each experimental dataset and additionally uses the **molecular formula** as input, which provides very strong prior information. For this reason, a direct numerical comparison is not fully fair or informative for isolating the contribution of our spectrum-to-structure framework.
> > >
> > > In addition, our previous robustness experiments with **Gaussian noise** and **baseline drift** support the same conclusion: MAST remains stable under significant spectral perturbations and consistently outperforms DiffSpectra in the noisy setting. Therefore, taken together, the new experimental evaluation and the noisy-spectrum results both support the robustness and practical usefulness of our method.
> > >
> > >
> > > > **Q2: Gating coefficients and gap between MAST (w/o Motif) and DiffSpectra..**
> > >
> > > **Reply:** Thank you for this helpful suggestion. A complete additional ablation on this point is unfortunately difficult to finish within the rebuttal period. However, we agree that the role of the gating mechanism should be clarified more explicitly.
> > >
> > > To better understand this effect, we inspected the learned gate values in our model. The gate coefficients have **a mean of 0.94** and **a standard deviation of 0.05**, indicating that the model assigns substantially larger weight to the **spectral condition** while using the **time embedding** mainly as auxiliary information. In other words, the conditioning in MAST is dominated by the spectrum signal rather than being equally mixed with the diffusion timestep information.
> > >
> > > This also partially explains the gap between **MAST (w/o Motif)** and **DiffSpectra**. Unlike our explicitly adaptive gating, **DiffSpectra** maps the spectrum through a simple MLP and directly **adds** it to the time embedding, without an explicit reweighting mechanism. This may make the spectral condition less effectively emphasized. We will add a clearer discussion of this point in the revised version.

---

### Official Review · Reviewer_AMrP · 2026-03-15

**Soundness:** 3
**Presentation:** 3
**Significance:** 3
**Originality:** 3
**Overall Recommendation:** 5
**Confidence:** 4

**Summary:**

This article presents a relevant issue. The authors claim to study a relevant challenge: molecular structure elucidation from spectra, especially the difficult one-to-many mapping from spectra to valid 2D/3D molecular structures. The proposed method, MAST, combines two main ideas: (1) a spectra-to-motif predictor that produces interpretable motif priors and injects them into a joint 2D–3D diffusion model, and (2) an MCTS-style tree search over checkpointed diffusion trajectories to improve test-time search efficiency and recovery under limited denoising budgets. The empirical results on QM9S are strong: the paper reports Acc@1 improving from 60.68% for DiffSpectra to 79.12% for MAST without MCTS and to 94.89% with MCTS, along with improvements on multiple similarity and 3D metrics. The motif prior and the search procedure are both supported by ablations.

**Compliance With Llm Reviewing Policy:**

Affirmed.

**Final Justification:**

The authors' rebuttal addressed my main concerns, i agree with that the score should be raised.

**Key Questions For Authors:**

1. Can the authors provide wall-clock runtime, GPU-hours, and memory comparisons for Best-of-N, Beam Search, and MCTS-guided diffusion at matched recovery levels, not just matched denoising-step budgets?
2. How sensitive are the results to the motif vocabulary design? What happens with fewer motifs, alternative vocabularies, or motifs extracted from a different corpus?
3.How much of the final gain comes from:
- motif conditioning in the generator,
- the spectra-structure reward model,
- the motif-consistency reward term,
- and the tree search itself?
4. Can the authors test on experimental spectra or at least a more realistic noisy/shifted benchmark to support claims of practical utility?
5. Are there examples where the motif prior is confidently wrong and actively harms search? A failure-case section would be valuable.

**Limitations:**

The paper would be significantly stronger with a more rigorous compute section. Report wall-clock, number of model evaluations, auxiliary reward-model cost, and maybe accuracy-versus-runtime curves.

Please add a clearer novelty discussion against prior diffusion + guided search work and against motif/intermediate-prediction approaches in spectra-to-structure tasks.

A stronger realism section would help a lot: noisy spectra, calibration under distribution shift, larger molecules, or at least a discussion of why QM9S should predict real-world performance.

I would also like a more detailed motif analysis: vocabulary construction, motif frequency distribution, ambiguity across motifs, and whether some motifs dominate the gains.

**Strengths And Weaknesses:**

## Strengths:

The paper is clearly motivated. It identifies two plausible bottlenecks in prior spectra-conditioned diffusion methods: over-reliance on a single global spectral embedding and expensive repeated full-trajectory sampling at inference. The method is reasonably well structured. The motif prior is interpretable and chemically meaningful in spirit, since it externalizes intermediate motif-level evidence rather than leaving all conditioning implicit in a single embedding. The checkpointed diffusion tree plus MCTS is also a coherent test-time scaling idea. The empirical gains are substantial and consistent across many metrics. Table 1 shows a very large jump in exact recovery and strong similarity scores, while Table 3 suggests both the motif prior and MCTS matter. Table 4 further supports that the motif predictor is accurate and that conditioning improves motif inclusion in generated molecules. I also appreciate that the paper discusses non-uniqueness and uncertainty in spectra-to-structure mapping, and mentions returning multiple candidates rather than one deterministic answer.

## Weaknesses/Concerns:

My biggest concern is external validity. The evaluation appears centered on QM9S, which is based on simulated spectra for small molecules. That makes it hard to judge whether the reported gains will transfer to realistic noisy experimental settings, broader chemical spaces, or more challenging molecular sizes and chemistries. The paper itself acknowledges this limitation in the conclusion and impact statement.

A second concern is fairness and transparency of the inference budget comparison. The paper argues that MCTS is more compute-efficient than repeated full-chain sampling and shows better exact recovery under denoising budgets, but the exact accounting of budget fairness is not fully convincing from the main paper alone. For example, the method uses auxiliary reward models, fast rollouts, tree width, and checkpointing; these choices may substantially affect wall-clock cost and memory, yet the main text mostly reports denoising-step budgets rather than true runtime or compute-normalized comparisons.

A third concern is novelty composition. The ingredients themselves are not individually very surprising: motif-like intermediate supervision is a familiar idea, and MCTS/test-time search over generative processes has prior precedent. The novelty is mainly in their integration for spectra-conditioned 2D/3D diffusion. That is still worthwhile, but the paper could better clarify what is conceptually new beyond combining known ideas in this domain.

---

> ### Author Rebuttal · Authors · 2026-03-29
>
> Thank you for your detailed and constructive feedback. We address your concerns below.
> > **W1: Limited evidence beyond QM9S.**
>
> **Reply:** We add two robustness evaluations.
>
> First, we perturb the input spectra with **Gaussian noise** and **baseline drift** to mimic realistic measurement errors. Results are shown in **MAST under noisy spectra** (https://anonymous.4open.science/r/2026-B258/MAST_Noise_Spectra.png).
>
> We also compare against **DiffSpectra** under combined perturbations, with results shown in **DiffSpectra vs. MAST under noisy spectra** (https://anonymous.4open.science/r/2026-B258/MAST_Diff_Noise.png).
>
> Second, we collect 25 **real experimental spectra** from the **NIST WebBook** for molecules, with results shown in **MAST on the real spectra** (https://anonymous.4open.science/r/2026-B258/Real_data.png).
>
> Overall, these results show that our model is robust: it remains stable under noisy perturbations and performs much better than DiffSpectra on both corrupted and real spectra. With **MCTS**, **Acc@1** further improves to **78.12%** under **SNR=30 dB + drift=5** and **61.76%** on the real subset.
>
> We further find that this robustness is supported by the **motif predictor**, whose performance drops only slightly under perturbations.
> > **W2: Efficiency comparison is incomplete without runtime/memory.**
>
> **Reply:** We additionally measure **wall-clock runtime per sample** and report **accuracy--runtime--memory** comparisons. See **accuracy-runtime-memory** (https://anonymous.4open.science/r/2026-B258/Acc-time_mem.png).
>
> The results show that, under comparable runtime, our method still achieves higher accuracy than Best-of-N and Beam Search. Our profiling also shows that denoising dominates the total cost. For example, under the **3000-step** MCTS setting, the **average number of simulations is only 48.2**. We also observe that **GPU memory usage is similar** across all methods because they share the same denoising model, while MCTS requires somewhat more **CPU memory** because it needs to maintain the search tree.
> > **W3: Novelty is mainly integration of known ideas.**
>
> **Reply:** Beyond integration, a key novelty is **adaptive guidance in conditional diffusion via tree search**. In spectra-conditioned diffusion, the best guidance strength is hard to choose and can vary across denoising stages: overly strong guidance may push samples off the data manifold, while weak guidance may be ineffective. By combining MCTS with guidance, our method can adaptively select better guidance across denoising stages.
> > **Q1: Can you provide runtime, GPU-hours, and memory comparisons?**
>
> **Reply:** Yes. We address runtime and memory in **W2**.
> > **Q2: How sensitive are the results to the motif vocabulary design?**
>
> **Reply:** We agree that motif vocabulary design is important. A complete comparison of  different vocabularies would require retraining both the **motif predictor** and the **diffusion model**, which is beyond the rebuttal period.
>
> Our current design is a trade-off between richer motif priors and reliable prediction. We find that perturbing the predicted motif prior causes clear performance drops, which suggests that the quality of motif prior matters. We will clarify this rationale and add the noisy-motif analysis in the revision.
> > **Q3: How much gain comes from motif conditioning, the scorer, motif reward, and tree search?**
>
> **Reply:** The gain from **motif conditioning** is already shown in **Table 3(A)**: adding the motif prior improves **Acc@1** from **69.94%** to **79.12%**. To further analyze the remaining gains, we add three ablations on **1024 samples**. Results are shown in:
> - **backup strategy** (https://anonymous.4open.science/r/2026-B258/backup.png)
> - **motif reward weight** (https://anonymous.4open.science/r/2026-B258/weight.png)
> - **reward scorer perturbation** (https://anonymous.4open.science/r/2026-B258/scorer.png).
>
> These results show that:
> - Search quality depends on the **backup strategy**, though all variants remain competitive
> - A **moderate motif-consistency reward weight** works best, while too little or too much hurts exact recovery
> - The full system benefits from a high-quality **spectra-structure scorer**, while remaining relatively stable under mild perturbation.
>
> > **Q4: Can you test on experimental or noisy spectra?**
>
> **Reply:** Yes. This is addressed in **W1**.
> > **Q5: Are there failure cases where the motif prior is confidently wrong?**
>
> **Reply:** Yes. We can identify such failure cases, although they are relatively infrequent.
> One representative example is as follows. For this molecule, the baseline top-1 prediction is correct:
> `[H]C([H])([H])C12N3C4([H])C([H])(C31[H])C42[H]`
>
> After introducing the motif prior, the top-1 prediction becomes incorrect:
> `[H]C([H])([H])C12N3C4([H])C3([H])C1([H])C42[H]`
>
> In this case, the motif predictor assigns very high confidence (**0.993**) to an absent motif `CC1CCNCC1` and the final prediction indeed contains this incorrect motif.

---

> > ### Author Rebuttal · Reviewer_AMrP · 2026-04-03
> >
> > My concerns are mostly resolved, and I will raise my score accordingly

---

> > > ### Author Response · Authors · 2026-04-07
> > >
> > > We thank the reviewer for the feedback and for acknowledging that the concerns have been addressed. We greatly appreciate your time and effort in providing these valuable suggestions, which have helped us strengthen the paper.

---

### Decision · Program_Chairs · 2026-04-30

**Decision:**

Accept (regular)

**Comment:**

This paper combines chemical motif priors with a diffusion model and MCTS search for spectra-conditioned 2D–3D molecular structure elucidation. Reviewers agree the work is technically sound and novel, and experimental results on the used QM9S are strong. One reviewer's concern is that only one dataset is used for experiments. However, I think new data curation may be out of the scope of this work. Given this, an acceptance is recommended. In the revision or future work, the authors may discuss or address the compute/latency limitations, and consider more datasets for this line of research.